



# The CryoGrid community model (version 1.0) - a multi-physics toolbox for climate-driven simulations in the terrestrial cryosphere

Sebastian Westermann[1,2], Thomas Ingeman-Nielsen[3], Johanna Scheer[3], Kristoffer Aalstad[1], Juditha Aga[1,2], Nitin Chaudhary[1,4], Bernd Etzelmüller[1], Simon Filhol[1], Andreas Kääb[1], Cas Renette[1], Louise Steffensen Schmidt[1], Thomas Vikhamar Schuler[1], Robin B. Zweigel[1,2], Léo Martin[5], Sarah Morard[6], Matan Ben-Asher[7], Michael Angelopoulos[8], Julia Boike[8], Brian Groenke[8,9], Frederieke Miesner[8], Jan Nitzbon[8], Paul Overduin[8], Simone M. Stuenzi[8], and Moritz Langer[8,10]

[1]Department of Geosciences, University of Oslo, Norway
[2]Center for Biogeochemistry of the Anthropocene, University of Oslo, Norway
[3]DTU Sustain, Technical Univeristy of Denmark, Kgs. Lyngby, Denmark
[4]Department of Physical Geography and Ecosystem Science, Lund University, Sweden
[5]Faculty of Geosciences, Utrecht University, Utrecht, The Netherlands
[6]Department of Geosciences, University of Fribourg, Fribourg, Switzerland
[7]EDYTEM Lab, Université Savoie Mont Blanc, CNRS, Le Bourget-du-Lac, France
[8]Permafrost Research Section, Alfred Wegener Institute Helmholtz Centre for Polar and Marine Research, Potsdam, Germany
[9]Department of Electrical Engineering and Computer Science, Technical University of Berlin, Berlin, Germany
[10]Geography Department, Humboldt-Universität zu Berlin, Berlin, Germany

**Correspondence:** Sebastian Westermann (sebastian.westermann@geo.uio.no)

**Abstract.** The CryoGrid community model is a flexible toolbox for simulating the ground thermal regime and the ice/water balance for permafrost and glaciers, extending a well-established suite of permafrost models (CryoGrid 1, 2 and 3). The CryoGrid community model can accommodate a wide variety of application scenarios, which is achieved by fully modular structures through object-oriented programming. Different model components, characterized by their process representations and parametrizations, are realized as classes (i.e. objects) in CryoGrid. Standardized communication protocols between these classes ensure that they can be stacked vertically. For example, the CryoGrid community model features several classes with different complexity for the seasonal snow cover which can be flexibly combined with a range of classes representing subsurface materials, each with their own set of process representations (e.g. soil with and without water balance, glacier ice).

We present the CryoGrid architecture as well as the model physics and defining equations for the different model classes, focusing on one-dimensional model configurations which can also interact with external heat and water reservoirs. We illustrate the wide variety of simulation capabilities for a site on Svalbard, with point-scale permafrost simulations using e.g. different soil freezing characteristics, drainage regimes and snow representations, as well as simulations for glacier mass balance and a shallow water body. The CryoGrid community model is not intended as a static model framework, but aims to provide developers with a flexible platform for efficient model development. In this study, we document both basic and advanced model functionalities to provide a baseline for the future development of novel cryosphere models.





# 1 Introduction

The terrestrial cryosphere is currently undergoing unprecedented changes, including thawing of permafrost, melting of glaciers and ice sheets, and changes in snow cover extent. In the last decade, permafrost temperatures have warmed almost everywhere in the circum-Arctic (Biskaborn et al., 2019), and the melting of excess ground ice can accelerate thawing in a positive feedback

loop, leading to the fast transformation of permafrost landscapes through thermokarst (Farquharson et al., 2019; Nitzbon et al., 2020; Turetsky et al., 2019). Glaciers worldwide have been retreating at increasing rates (e.g. Hugonnet et al., 2021; Huss and Hock, 2018), and are important contributors to global sea-level rise. On regional scales, glacier retreat can affect e.g. freshwater availability, infrastructure and wildlife (e.g. Kaser et al., 2010).

Cryosphere land surface models are important tools to investigate the sensitivity of the terrestrial Cryosphere under complex

environmental and climatic conditions. In particular, the models allow us to project climate change impacts and thus answer urgent questions on the future of the cryosphere. As an example, glacier mass balance models are important tools for estimating the response of ice masses to a changing climate. They aid the investigation of the current state of the cryosphere in areas where in-situ observations are hard to obtain and can be used to estimate the past and future evolution of glaciers (e.g. Mankoff et al., 2021; Schmidt et al., 2020; van Pelt et al., 2021). Similarly, numerical models are highly important to investigate the current

state of permafrost, in particular since permafrost is usually not visible at the Earth's surface and only a limited number of measurement sites exist. Permafrost models provide insights into the evolution of Arctic landscapes and help us understand how these fragile ecosystems respond to natural and human-caused disturbances.

The purpose of land surface models is to describe nature in an adequate way, which means that the models must reproduce observations of the targeted physical parameters. This can be achieved by models of different complexity, from simple semi-

empirical models trained by observations to physically-based schemes which run independent of observations. Examples of semi-empirical models are degree day melt models for glacier mass balance (Gabbi et al., 2014; Reveillet et al., 2017), or the TTOP equilibrium model to estimate permafrost temperatures (Smith and Riseborough, 1996), which both are designed for a certain application. In contrast, physically-based land surface models can simulate both glacier mass balance and permafrost thermal regime with the same model framework, relying on universal formulations, such as the surface energy balance and

Fourier's Law of heat conduction. Over the past decades, land surface models have grown in complexity to incorporate a wide range of processes from various disciplines, such as biophysics, biogeochemistry, hydrology, or ecology (Fisher and Koven, 2020). In theory, continuous improvements over time could eventually lead to a unified "land surface model of everywhere, everything and all times" (Blair et al., 2019), which can reproduce and explain observations of all land surface variables, irrespective of their spatial and temporal scales. In reality, however, complex land surface models feature a large number

of model parameters whose variations in space and time are poorly constrained. This severely compromises their advantage over simpler model approaches in many use scenarios, in addition to strongly increased computation demands. Therefore, simple, less process-rich models have significant advantages in many practical applications and are typically employed for high-resolution (e.g. Obu et al., 2019), long time-scale and/or large ensemble simulations.





The CryoGrid suite of permafrost models have provided three model categories of increasing level of complexity to conduct
a wide range of permafrost studies. **CryoGrid 1** is an equilibrium model to compute mean annual ground temperatures at the
top of the permafrost table (TTOP) as the only output (Gisnås et al., 2013), which in particular makes it possible to infer the
presence/absence of permafrost. It relies on surface or air temperatures as input, in addition to $n$-factors to parameterize the
effects of snow cover and active layer properties on the seasonal heat exchange. While CryoGrid 1 was used for fine-scale
process studies (Gisnås et al., 2014, 2016), its main use was for large-scale mapping of permafrost extent and temperatures,
e.g. the generation of a permafrost map for Scandinavia (Gisnås et al., 2017). By using globally available remote sensing and
reanalysis (MODIS land surface temperature, ERA reanalysis) to force CryoGrid 1, permafrost maps on continental scale could
be produced (Westermann et al., 2015). Later, this processing chain was extended to produce 1km-resolution permafrost maps
of the Northern hemisphere (Obu et al., 2019) and Antarctica (Obu et al., 2020). Due to a low number of parameters and an
efficient and simple implementation, CryoGrid 1 allowed for large-scale ensemble simulations at $1\,\mathrm{km}$ gridcell size, so that
the effect of small-scale spatial variability of snow depths and ground properties on the thermal regime could be represented
statistically. Similar modeling approaches building on analytical formulations for ground temperature and active layer thickness
have also been applied at regional scales (e.g. GIPL1 in Alaska, Sazonova and Romanovsky, 2003).

Being an equilibrium model, CryoGrid 1 is generally not well suited for climate change simulations, missing transient
processes. For instance, changes of the model forcing impact ground temperatures without time delay in the this equilibrium
model, when in reality subsurface processes like ground ice melt delay the response of ground temperature. Therefore, the
transient model **CryoGrid 2** is employed for mapping climate change impacts, using similar spatially distributed forcing data
sets as CryoGrid 1 (Czekirda et al., 2019; Westermann et al., 2013, 2017). Similar to the GIPL2 model (Jafarov et al., 2012),
CryoGrid 2 computes ground temperatures from conductive heat transfer through the ground and the snowpack, validated in
permafrost regions of northern Siberia and Norway (Beermann et al., 2017; Langer et al., 2013; Westermann et al., 2011).
Due to its relative computational efficiency, CryoGrid 2 has also been adapted to simulate multi-millennial paleo-permafrost
evolution, for example during deglaciation on Iceland (Etzelmüller et al., 2020). CryoGrid 2 was also used to model the
evolution of permafrost beneath the circum-Arctic continental shelves (Overduin et al., 2019) with model forcing computed
for the last $450\,\mathrm{kyr}$ from model reconstructions of glaciation, sea level and air temperature. The implementation of coupled
heat and salt diffusion equations in offshore sediments showed mitigation of seabed seasonal freezing and enhancement of
subsea permafrost degradation rate because of the presence of salt (Angelopoulos et al., 2019). Further applications of the
model to thermokarst lagoon and coastal settings demonstrated how brine rejection lowers sediment freezing temperature and
slows the refreezing of thawed sediments (Angelopoulos et al., 2020, 2021).

**CryoGrid3** is a land surface model that accounts for land atmosphere coupling by simulating the surface energy balance,
similar to the COUP (e.g. Marmy et al., 2013), GEOtop (Rigon et al., 2006) and SURFEX (Barrere et al., 2017) models. Cryo-
Grid 3 features a representation of excess ground ice, so that ground subsidence and thermokarst pond formation upon thaw
can be simulated (Westermann et al., 2016). Furthermore, it was used to simulate heat transport in water bodies as well as their
impact on the thermal regime and the thaw threshold of the permafrost below (Langer et al., 2016).The implementation of the
state-of-the-art snow scheme Crocus (Vionnet et al., 2012) into CryoGrid 3 allowed for transient representation of internal snow





properties as well as wind redistribution of snow, which was key to realistically simulate local ground temperature dynamics
in snow-rich regions (Zweigel et al., 2021). As about 55% of permafrost area is covered by boreal forest, CryoGrid 3 was
extended by a multilayer vegetation scheme (Bonan et al., 2018) for the modeling of the thermal and hydrological permafrost
conditions under boreal forest covers (Stuenzi et al., 2021b, a). CryoGrid 3 was further extended by a bucket hydrology scheme
for unfrozen conditions, as well as lateral transport of water, heat and snow; this version has been evaluated and applied for
different permafrost ecosystems (Martin et al., 2019; Nitzbon et al., 2019). Nitzbon et al. (2020), Nitzbon et al. (2021) and
Martin et al. (2021) further demonstrated the applicability of CryoGrid 3 to simulate complex permafrost landscape evolution
over a range of spatial (plot-to-landscape) and temporal (years-to-centuries) scales. Such spatially distributed realizations of
CryoGrid 3 (denoted "laterally coupled tiles") aim for a three-dimensional representation of permafrost hydrology, similar to
cold-region hydrological models, such as WASIM (Debolskiy et al., 2021), TopoFlow (Schramm et al., 2007), SUTRA-Ice
(McKenzie et al., 2007), PFLOTRAN-Ice (Karra et al., 2014) and Amanzi-ATS (Abolt et al., 2020).

While CryoGrid 1, 2 and 3 are partly based on the same model formulations and process parametrizations, they are essentially
different models regarding numerics and code structure. Furthermore, they have been adapted for many different use cases
creating numerous derivatives of slightly different model versions that are not necessarily compatible.

In this study, we present the architecture of a new CryoGrid community model, which comprises most of the functionalities
demonstrated in CryoGrid 1-3, while going beyond in many aspects. In particular, the CryoGrid community model is not a
single model, but a modular collection of models with different functionalities which can be combined with each other to fit the
requirements of a variety of applications. We describe key aspects of the model physics for one-dimensional simulations, espe-
cially when going beyond the capabilities of previously documented CryoGrid 1-3. An example is a new glacier mass balance
module which extends the capabilities of the CryoGrid community model beyond permafrost. We showcase the possibilities
of this new simulation tool with point simulations for Svalbard, as well as benchmark simulations against analytical solutions
and reference experiments.

## 2 CryoGrid community model description

### 2.1 Architecture and setup

#### 2.1.1 Model concept - modularity through object-oriented programming

The CryoGrid community model is based on an object-oriented programming paradigm implemented in the programming
language Matlab in which "objects" are referred to as "classes". A class is a defined structure, which consists of a class-
specific set of variables, as well as class-specific functions to modify these variables. A variable within a class can once
again be a class (more precisely a pointer to another class), typically of a different type, which makes it possible to create a
tree-like structure with different hierarchical levels (Fig. 1). Hereby, each class at a given hierarchy level contains pointers to
the lower level classes, with different levels representing different functionalities within the simulation system. In CryoGrid,
each functionality level is represented by specific class types for which typically several options (i.e. different classes) are



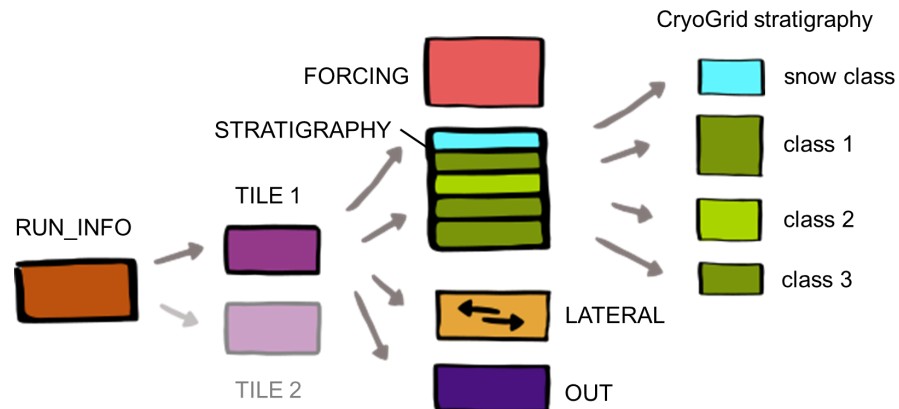

**Figure 1.** Hierarchy (left to right) of the different types of CryoGrid classes required for multi-physics simulations within the CryoGrid community model, as described in this study. Arrows represent pointers to classes employed in the hierarchical level below. Other hierarchies, potentially comprising additional and different types of classes, are fully possible within the CryoGrid community model and can be implemented in the future, e.g. a set of classes for spatially distributed applications with less customizable, but computationally more efficient simulation tools (Sect. 4.6). See text. The CryoGrid stratigraphy is depicted in more detail in Fig. 2.

available, allowing to customize and adapt the model setup. Classes of a given type feature mandatory variables and functions with standardized inputs and outputs, so that they become interchangeable building blocks of a modular simulation system with intrinsic compatibility. For each class type, the best-fitting class for the particular use case must be selected by the user. Furthermore, each class has specific parameters which must be set by the user and further control the behavior of the model system. A description of all available CryoGrid classes employed in this study is presented in Suppl. 2.

Fig. 1 depicts the class types and associated hierarchy that are employed to realize the multi-physics simulations described in the remainder of this study. However, it is possible to implement other configurations (e.g. with different class types and hierarchies) within the CryoGrid community model (see Sect. 4.6 for a discussion of such possibilities). The "RUN_INFO" class is the only mandatory class type, which represents the uppermost level of any class hierarchy. For the multi-physics simulations described in this work, the second level in the class hierarchy uses the class type "TILE" (Fig. 1). The purpose of a TILE class is to perform a classic one-dimensional model simulation over a predefined time period with a predefined forcing dataset (as in e.g. CryoGrid 2 and 3). RUN_INFO classes, on the other hand, organize the model simulations. Depending on which RUN_INFO class is selected, they can for example launch only a single TILE class, or several TILE classes (e.g. several independent simulations, representing grid cells, ensemble members, etc.) either sequentially or in parallel. Additionally, model spin-up can be implemented by sequentially simulating a number of TILE classes. For example, one TILE classes can be used for the spin-up phase and another for the target period of the simulations, which is initialized by the model state of the spin-up TILE class (see Sect. 3.1.4 for an accelerated spin-up procedure using a sequence of TILE classes). A description of all RUN_INFO classes employed in this study is provided in Suppl. 2.





The multi-physics simulations described in this work all employ the TILE class "TILE_1D_STANDARD" (see Suppl. 2 for
more details) that performs a full model simulation, from model initialization to the generation of the model output. For this
purpose, a range of specialized class types are employed (hierarchy level 3 in Fig. 1), which control different aspects of the
simulation, such as the initialization of model state variables ("STRAT_STATVAR" classes), the model forcing ("FORCING"
classes), and the model output ("OUT" classes). We do not describe the entire functionality of these class types (see Suppl.
2 for details), but only provide a few examples showcasing the modularity. FORCING classes, once again adhering to the
strict protocol of mandatory variables and internal functions, are designed to provide the required model forcing at a specific
timestep. Different FORCING classes are available, for example a class simply interpolating the raw model forcing, and a
class reprojecting the radiative components of the raw model forcing based on slope and aspect. The choice of the OUT class
determines what kind of and how model output is stored. For development and testing, a class storing the entire variable space
can be used, while users may want to design a purpose-built OUT class which only stores their model variables of interest.
There is also an OUT class storing the full model state after the final simulation timestep, which can be used to start a new
simulation based on that state (i.e. initialize a new TILE class). STRAT_STATVAR classes (abbreviation for "stratigraphy of
state variables", not shown in Fig. 1) are employed to calculate depth profiles of model variables to define the initial state on the
model grid. Depending on the class, these can be provided as layers with constant values, or by interpolation between values
at defined depths.

The backbone of the modularity within TILE_1D_STANDARD is the possibility to define a vertical stack of classes, each
employing different model physics and parameterizations within layers of the model domain (see Fig. 2, Sects. 2.1.2, 2.2). In
the following, we refer to this vertical stack of classes as "CryoGrid stratigraphy", and the classes within this stack, which
encode the model physics, as "stratigraphy classes". The vertical domain covered by each of the stratigraphy classes (i.e. the
stratigraphy of these classes) is again assigned via a purpose-build class ("STRAT_CLASSES", denoted "STRATIGRAPHY"
in Fig. 1). Finally, TILE_1D_STANDARD features a class type controlling lateral interactions with an external environment
(denoted "LATERAL" in Fig. 1). These LATERAL classes are described in Sect. 2.3.

### 2.1.2 Multi-physics representation with stratigraphy classes

A one-dimensional model simulation (in TILE_1D_STANDARD, see above) is realized by vertically stacking different stratig-
raphy classes (Fig. 2) which are each defined by their specific model physics and state variables. Examples of stratigraphy
classes are ground columns with and without water balance, water bodies, glaciers and snow with different levels of process
representation (see Sect. 2.2 for details). Each class occupies a certain vertical domain, with the boundary conditions applied
to the upper- and lowermost classes.

Within each stratigraphy class, CryoGrid computes the time evolution for state variables, such as ground temperature and
water/ice contents. We distinguish between prognostic state variables for which a time derivative is calculated and which are
then integrated in time to advance to the next timestep, and diagnostic state variables which are not time-integrated. For the
prognostic state variables, CryoGrid uses the simple time integration scheme "first-order forward Euler", i.e. the new model
state is computed as the old model state plus time derivatives times the model timestep (see Sect. 2.2.9 for details). When





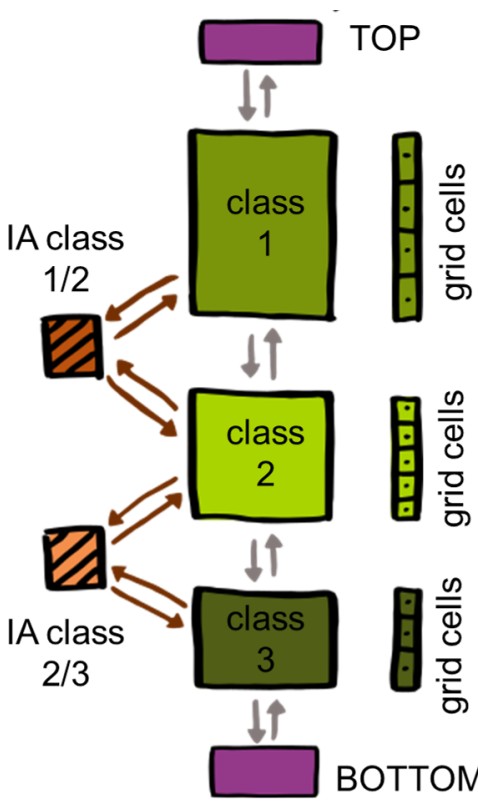

**Figure 2.** Example of a CryoGrid stratigraphy in the tile class TILE_1D_standard, consisting of the stratigraphy classes 1 to 3 coupled by interaction (IA) classes that specify the exchange of heat and mass between pairs of stratigraphy classes. The stratigraphy is realized as a linked list, with pointers between classes and interaction classes (symbolized by arrows), and the top and bottom of the list represented by a dedicated TOP and BOTTOM class (which have no other functionality). Each stratigraphy class has its own model grid and state variables.

the new model state of the prognostic state variables is obtained, diagnostic state variables are calculated from the prognostic variable by constitutional relationships. This calculation is instantaneous, i.e. does not depend on the employed timestep. It is

possible that a physical property (e.g. temperature) is a prognostic variable in some stratigraphy classes and diagnostic in others. Stability and accuracy of the time integration is ensured by automatically selecting an appropriate timestep. The calculation of a suitable timestep is not accomplished by well-known stability criteria for the first-order forward Euler scheme, but explicitly takes the physics represented by the individual stratigraphy classes into account to ensure both a stable and accurate simulation (see Sect. 2.2.9).

Interactions between stratigraphy classes are realized by interaction classes (Fig. 2) that compute fluxes across the boundaries between pairs of stratigraphy classes. Thus, compatibility of two stratigraphy classes (i.e. if they can border each other in the CryoGrid stratigraphy) can be ensured by providing a dedicated interaction class. Note that the compatibility depends on





the order of the two classes in the stratigraphy. In particular, interaction classes compute fluxes across boundaries between stratigraphy classes which are required for computing time derivatives of state variables in the prognostic step of the time

integration. For example, if two stratigraphy classes have temperature as state variables and the two modules are connected through heat conduction, the interaction class computes the conductive heat flux between the adjacent grid cells of the classes. If two stratigraphy classes feature different state variables, the interaction class must contain the necessary code to compute the correct fluxes for both involved classes. For example, if only one of the classes is hydrologically active, while the sum of water and ice contents are static for the other class (see Sect. 2.2), the interaction class must provide a zero water flux boundary

condition for the hydrologically active class to reflect the fact that water flow through the boundary is not possible. During initialization of the CryoGrid stratigraphy, the correct interaction class is automatically selected for each pair of stratigraphy classes.

### 2.1.3  Dynamic behavior with stratigraphy class triggers

An important feature of stratigraphy classes is their ability to modify or rearrange the CryoGrid stratigraphy itself, if a certain

condition (referred to as "trigger") is met. In particular, a class can remove itself from the CryoGrid stratigraphy, or insert a new stratigraphy class above its own position. As an example, a dynamic representation of ponds (using water body and excess ice classes, see Sect. 2.2) can be achieved by such triggers which modify the CryoGrid stratigraphy. When surface water pooling up over initially dry ground reaches a user-defined threshold, a water body class representing the physics of energy transfer within a water body is created and inserted in the CryoGrid stratigraphy. Likewise, if the water depth of a water body

drops below that threshold, the water body class is automatically removed. In this process, all state variables are automatically adjusted to ensure mass and energy conservation.

A special situation is the representation of the seasonal snow cover which again is handled by stratigraphy class triggers creating a snow class (Sect. 2.2.6) upon initial snowfall and removing it when all snow has melted. For the numerical scheme, handling a very shallow initial snow cover poses significant problems, as this results in a small grid cell size and thus very

small timesteps. Therefore, snow classes are attached and detached in two stages in the CryoGrid community model. After the first snowfall, the snow class is added as a so-called "CHILD" to the uppermost stratigraphy class (Fig. 3a), i.e. it is not part of the CryoGrid stratigraphy, but evolves as part of the uppermost stratigraphy class. In this CHILD state, the snow is assumed to cover only a fraction of the uppermost stratigraphy class, with the upper boundary condition applied to both classes and fluxes weighed by their aerial coverage. This way, it is possible to assign the snow cover a sufficient thickness to prevent numerical

problems; as more snow accumulates, the snow class expands its aerial coverage. Finally, when the amount of snow is sufficient to be handled without numerical problems, the snow class and the associated interaction class are simply rearranged, so that the snow class becomes part of the CryoGrid stratigraphy (Fig. 3b) and thus cover its full areas. The procedure is mirrored upon snowmelt, with the snow class first becoming a CHILD and finally being removed completely upon completion of melt.





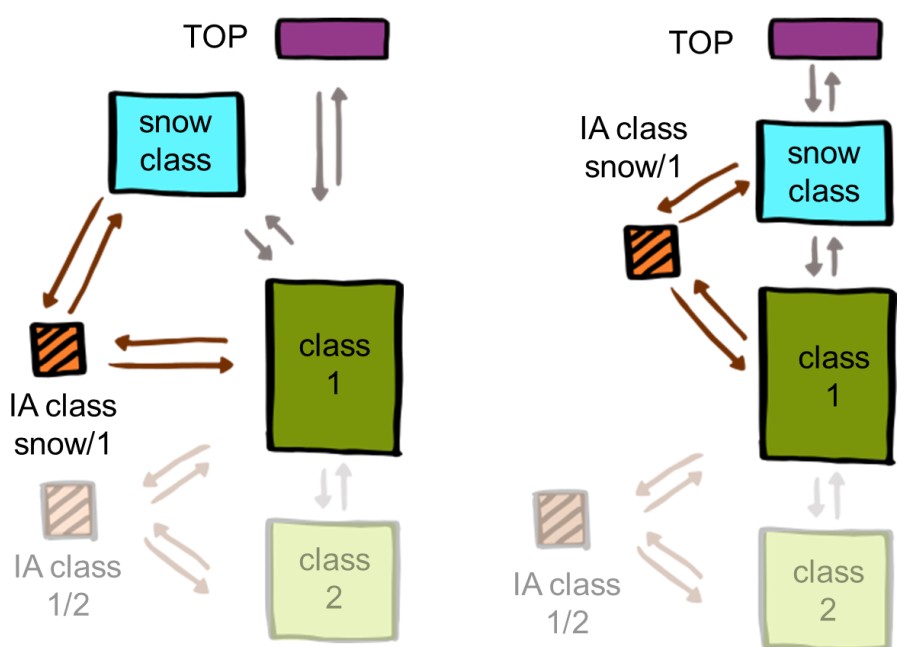

**Figure 3.** Schematic representation of the CryoGrid stratigraphy when a snow cover is present (represented by dedicated snow classes, Sect. 2.2.6). Left: snow is initially added as a so-called CHILD to the uppermost subsurface class (class 1), i.e. it is not part of the regular stratigraphy, but is addressed by specific pointers. In the CHILD phase, the snow is assumed to only cover a part of the surface area, e.g. the surface energy balance is calculated as a mix of snow-covered and snow-free ground. Right: when the snow water equivalent exceeds a user-defined threshold, the stratigraphy is rearranged and the snow class becomes the uppermost class of the regular stratigraphy. The process is reversed when the snow melts.

### 2.1.4 Model operation

All CryoGrid simulations are controlled by a parameter file which defines all aspects of the run, such as the definition of the CryoGrid stratigraphy and lateral interactions, the forcing data, the model grid and the model output format. At this point, the parameter file can either be set up as a spreadsheet (MS Excel or compatible programs) or as a text file in YAML format.

   In the parameter files, all classes required for the simulation are defined, in no particular order. Each class is identified by its name and a unique index which makes it possible to define the same class several times with different parameters.

Furthermore, all mandatory parameters specific to each of the classes must be specified. In the hierarchy of the CryoGrid classes (Fig. 1), the classes on the following level are defined as parameters in the classes of the previous level. Upon initialization, the uppermost hierarchy level (defined as the RUN_INFO class with index 1) is read first, which provides the information for reading the classes in the second level, and so on. In this process, the class connections by pointers are established. Note that the standardized class structure with mandatory variables and functions facilitates a generalized initialization routine which

does not make use of specific knowledge of the involved CryoGrid classes and their hierarchy.





## 2.2 Physics and defining equations of stratigraphy classes

At present, there are around ten stratigraphy classes, each with different defining equations and model physics which generally contain additional parameters and options to customize its behavior. However, the classes share many common parts and features. In the following, we describe the defining equations and parameterizations of the different model components and

categories. A description of each stratigraphy class is provided in Supplement 2.

### 2.2.1 State variables and model grid

All stratigraphy classes (except the equilibrium TTOP model class, Sect. 2.2.2) feature variables for the subsurface properties volumetric mineral content $\theta_m$, volumetric organic content $\theta_o$, volumetric water content $\theta_w$ and volumetric ice content $\theta_i$. A depth stratigraphy of mineral and organic contents is generally provided by the user which also defines the porosity as

$\phi = 1 - \theta_m - \theta_o$. In some stratigraphy classes, the sum of water and ice contents $\theta_{wi} = \theta_w + \theta_i$ is constant and provided by the user, while it evolves dynamically in others, driven by precipitation, evapotranspiration (Sect. 2.2.4) and potentially lateral runoff (Sect. 2.3.2). Finally, there is an air phase $\theta_a$ defined by $\theta_a + \theta_{wi} = \phi$. Each model grid cell has an enthalpy state, $e$ [$\mathrm{J\,m^{-3}}$], which is composed of a sensible and a latent part:

$$e(T, \theta_w) = cT - L_{\mathrm{sl}}^{\mathrm{vol}}(\theta_{wi} - \theta_w), \tag{1}$$

where $c$ [$\mathrm{J\,K^{-1}\,m^3}$] is the volumetric heat capacity, $T$ [°C] the temperature, and $L_{sl}^{vol}$ the volumetric latent heat of water freezing [$\mathrm{J\,m^{-3}}$]. The zero point of the enthalpy is thus defined as $T = 0°C$ and $\theta_i = 0$, i.e. the grid cell is at $0\,°C$, but all water is unfrozen. While $L_{sl}^{vol}$ is a constant, $c$ is computed from $\theta_m$, $\theta_o$ and $\theta_{wi}$ and the specific volumetric heat capacities of the mineral, organic, water and ice phases ($c_m$, $c_o$, $c_w$, $c_i$) as:

$$c = \begin{cases} \theta_m c_m + \theta_o c_o + \theta_{wi} c_i & \text{for } T < 0\,°C \\ \theta_m c_m + \theta_o c_o + \theta_{wi} c_w & \text{for } T \geq 0\,°C \end{cases} \tag{2}$$

Note that the heat capacity of a potential air phase is neglected. While these state variables are employed in the stratigraphy classes described in this study, their use is not mandatory within the CryoGrid community model and fully valid stratigraphy classes with different sets of state variables can be created. The compatibility with existing stratigraphy classes is ensured by appropriate interaction classes (Sect. 2.1.2) which compute fluxes between classes and, if necessary, convert between different sets of state variables.

The model grid is defined by the user, again using a dedicated "GRID" class. At present, only one grid class is implemented, in which constant grid cell sizes are specified within a sequence of layers. Typically, the smallest grid cell size is defined for the top layer and the largest for the bottom layer. Other grid classes, e.g. with grid cell sizes increasing logarithmically with depth, could be implemented in a straight-forward way.





### 2.2.2 Coupling to model forcing and boundary conditions of model domain

At the uppermost stratigraphy class of the CryoGrid stratigraphy (Fig. 2), the upper boundary condition is applied which simulates the coupling to the model forcing. Three different schemes are implemented at this point, broadly providing the functionality of the CryoGrid 1, 2 and 3 models.

*Equilibrium TTOP approach*: Used within CryoGrid 1, the TTOP approach offers an efficient way to estimate mean annual ground temperature ($MAGT$) directly from the model forcing. For this purpose, freezing and thawing degree days at the surface ($FDD_s$ and $TDD_s$) are calculated from the temperature forcing (often using air temperature to approximate surface temperature) and semi-empirical $n$-factors which phenomenologically simulate the asymmetry of heat transfer in the ground between freezing and thawing periods:

$$MAGT = \begin{cases} \frac{1}{\tau}(n_f FDD_s + r_k n_t TDD_s) & \text{for } n_f FDD_s + r_k n_t TDD_s \leq 0 \\ \frac{1}{\tau}(\frac{1}{r_k} n_f FDD_s + n_t TDD_s) & \text{for } n_f FDD_s + r_k n_t TDD_s > 0 \,, \end{cases} \tag{3}$$

with $\tau$ the number of days in the period for which the TTOP model is applied. Setting $n_f$ and $n_t$ unlike unity causes a temperature offset between the model forcing and the ground surface related to processes during the frozen season (in particular caused by the insulating snow cover) and the thawed season (e.g. caused by incoming radiation modified by slope and aspect). In the same fashion, $r_k$ causes a temperature offset between the ground surface and the top of the permafrost due to differences in active layer thermal conductivities and thus heat transfer between summer and winter. Detailed derivations of the TTOP equation (Eq. 3) are presented in Romanovsky and Osterkamp (1995) and Westermann et al. (2015). The TTOP approach calculates the ground temperature in equilibrium with the applied model forcing, i.e. the ground temperature that would eventually be reached if the model forcing was repeatedly applied for an infinite time period. The approach is not suited to capture ground temperatures during periods of rapid change, especially when an insulating top layer and ground ice delays the penetration of the surface temperature signal into the ground. For this reason, the TTOP approach should preferably be applied to longer time periods, e.g. for one or several decades of model forcing. Moreover, only a single temperature value is delivered without specifying a depth, or even a depth profile of ground temperatures. However, the TTOP model class can be vertically coupled to stratigraphy classes simulating heat conduction into the ground (Sect. 2.2.3), effectively providing a modified, temperature boundary condition (see below) which accounts for temperature offsets caused by the snow cover and active layer dynamics. Such model setups are particularly useful for simulating temperature dynamics in deeper layers for long timescales, as they do not need to resolve the seasonal freeze-thaw cycle, making them very efficient computationally.

*Temperature boundary condition*: Transient simulations with heat conduction-based models (Sect. 2.2.3) require specification of boundary conditions, which can be either time ($t$) series of temperature, $T_{\text{ub}}(t)$, or time series of the energy flux into the first model grid cell, $F_{\text{ub}}(t)$ [$\text{W m}^{-2}$]. In the CryoGrid community model, it is possible to specify a temperature boundary condition (as in the CryoGrid 2), which for each time $t$ is translated to a heat flux into the first model grid cell as

$$F_{\text{ub}}(t) = K_{h,1}(t) \frac{T_{\text{ub}}(t) - T_1(t)}{\Delta z_1} \tag{4}$$





where $K_{h,1}$ [$\mathrm{W\,m^{-1}\,K^{-1}}$] denotes the thermal conductivity (Sect. 2.2.3), $T_1$ the temperature and $\Delta z_1$ the thickness of the uppermost model grid cell. Note that Eq. (4) assumes that $T_{\mathrm{ub}}(t)$ is assigned to a virtual grid cell of thickness $\Delta z_1$ above the uppermost grid cell. Air temperatures are most commonly used for $T_{\mathrm{ub}}(t)$, but this can be adapted by selecting (and if necessary modifying) an appropriate FORCING class (Sect. 2.1.1) for the simulations.

*Surface energy balance*: The energy flux into the first model grid cell, $F_{\mathrm{ub}}(t)$, can also be calculated from the surface energy balance (SEB), largely similar to the implementation in CryoGrid 3. The required forcing data include incoming short- and longwave radiation ($S_{\mathrm{in}}$ and $L_{\mathrm{in}}$) [$\mathrm{W\,m^{-2}}$], precipitation ($P_s$, solid and $P_l$, liquid) [$\mathrm{mm\,d^{-1}}$] and air pressure $p$ [Pa], as well as air temperature $T_{\mathrm{air}}$, wind speed $U$ [$\mathrm{m\,s^{-1}}$] and specific humidity $q_{\mathrm{air}}$ [kg water vapor/kg air] at height above ground $h$ [m] (e.g. Westermann et al., 2016). $F_{\mathrm{ub}}$ is then calculated from the surface energy balance equation.

$$F_{\mathrm{ub}}(t) = S_{\mathrm{in}}(t) - S_{\mathrm{out}}(t) + L_{\mathrm{in}}(t) - L_{\mathrm{out}}(t) - Q_h(t) - Q_e(t), \tag{5}$$

where $S_{\mathrm{out}}$ and $L_{\mathrm{out}}$ denote outgoing short- and longwave radiation (both defined positive) and $Q_h$ and $Qe$ the sensible and latent heat flux (both defined positive when cooling the surface). Note that the ground heat flux $Q_g$ is not explicitly represented, but becomes manifest in both an enthalpy change of the uppermost grid cell and a conductive heat flux between the two uppermost grid cells. It can be calculated by the user as $Q_g = F_{\mathrm{ub}}A$ (assuming the same sign convention as for sensible and latent heat flux). The outgoing shortwave radiation is computed with the surface albedo $\alpha_s$

$$S_{\mathrm{out}} = \alpha_s S_{\mathrm{in}} \tag{6}$$

In most classes, a single broad-band (i.e. spectrally averaged) albedo is employed, but some classes resolve several spectral bands for which the albedo can vary. Furthermore, a constant albedo is provided by the user in some classes, while the albedo is parameterized as a function of model state variables in others. The outgoing longwave radiation is computed from Stefan-Boltzmann and Kirchhoff's Law as

$$L_{\mathrm{out}} = \epsilon\sigma_{\mathrm{sb}}(T + T_{\mathrm{mfw}})^4 + (1-\epsilon)L_{\mathrm{in}}, \tag{7}$$

with $\sigma_{\mathrm{sb}}$ the Stefan-Boltzmann constant [$\mathrm{kg\,m^{-2}\,K^{-4}}$], $T_{\mathrm{mfw}} = 273.15\,\mathrm{K}$ the freezing temperature of free water and $\epsilon$ [-] the surface emissivity.

The sensible heat flux is computed from air temperature at defined height above ground $h$ and the temperature of the first grid cell as

$$Q_h = -\frac{\rho_{\mathrm{air}}c_p}{r_a}(T_{\mathrm{air}} - T_1), \tag{8}$$

with $\rho_{\mathrm{air}}$ air density [$\mathrm{kg\,m^{-3}}$] and $c_p$ the air heat capacity [$\mathrm{J\,kg^{-1}\,K^{-1}}$] at constant pressure. The aerodynamic resistance $r_a$ is calculated from Monin-Obukhov similarity theory (Monin and Obukhov, 1954), as in CryoGrid 3, with

$$r_a = \frac{1}{\kappa^2 U}\left(\ln\frac{h}{z_0} - \psi_{\mathrm{M}}(\frac{h}{L_*}, \frac{z_0}{L_*})\right)\left(\ln\frac{h}{z_0} - \psi_{\mathrm{H,W}}(\frac{h}{L_*}, \frac{z_0}{L_*})\right). \tag{9}$$

Here, $U$ is the wind speed at height $h$ above ground, $\kappa = 0.4$ the von Kármán constant, $z_0$ [m] the roughness length (assumed equal for heat, water and momentum) and $\psi_{\mathrm{M}}$ and $\psi_{\mathrm{H,W}}$ integrated atmospheric stability functions, equal to the ones used in





CryoGrid 3 (Westermann et al., 2016). The Obukhov length $L_*$ [m] is calculated from the sensible and latent heat flux values in the same way as CryoGrid 3, using the flux values computed for the previous timestep (see Westermann et al., 2016).

The latent heat flux is calculated as

$$Q_e = -\rho_{\text{air}} L_{\text{lg,sg}} \frac{f}{r_e} (q_{\text{air}} - q_1), \tag{10}$$

where $L_{\text{lg,sg}}$ [J kg$^{-1}$] are the latent heat of evaporation (lg) and sublimation (sg), which are employed for $T_1 \geq 0\,°\text{C}$ and $T_1 < 0\,°\text{C}$, respectively. Depending on the subsurface class used, different formulation for the reduction factor $f$ [-], the resistance $r_e$ [s m$^{-2}$] and the specific humidity above the surface $q_1$ are employed. Four schemes can broadly be distinguished:

1. For subsurface classes with unlimited surface water or ice supply (e.g. classes representing snow cover or a water body), $r_e = r_a$, $f = 1$ and $q_1 = q_{\text{sat}}(T_1)$, i.e. the specific humidity at saturation for the temperature of the uppermost grid cell,
calculated with the Magnus equation. The evaporation thus corresponds to the potential evaporation.

2. For subsurface classes without water balance (i.e. water plus ice content constant in time), $f = 1$, $q_1 = q_{\text{sat}}(T_1)$ and $r_e = r_a + r_s$, with $r_s$ [s m$^{-1}$] a user-defined surface resistance to evaporation (see Westermann et al., 2016).

3. For subsurface classes with bucket water scheme (Sect. 2.2.4), the potential evapotranspiration (i.e. $r_e = r_a$ and $q_1 = q_{\text{sat}}(T_1)$) is multiplied by the reduction factor $f$ taking soil water availability into account. For unfrozen ground, $f$ is
calculated from the water availability coefficients of soil grid cells i, $\alpha_i^\theta$, as

$$f = f_{\text{tr}} \sum_i \alpha_i^\theta \Delta z_i e^{-d_i/d_{\text{tr}}} / \sum_i \Delta z_i e^{-d_i/d_{\text{tr}}} + f_{\text{ev}} \sum_i \alpha_i^\theta \Delta z_i e^{-d_i/d_{\text{ev}}} / \sum_i \Delta z_i e^{-d_i/d_{\text{ev}}}, \tag{11}$$

which allows the user to specify the partitioning in transpiration (fraction $f_{\text{tr}}$) and evaporation (fraction $f_{\text{ev}} = 1 - f_{\text{tr}}$) with different characteristic depth $d_{\text{tr}}$ and $d_{\text{ev}}$. Both evaporation and transpiration are assumed to decay exponentially with depth $d_i$ below the surface (taken positive). Furthermore, the weighting of each grid cell depends on the thickness of grid cell i, $\Delta z_i$, and the water availability coefficient calculated with the user-defined field capacity $\theta_{\text{fc},i}$ [-] as $\alpha_i^\theta =$
$0.25(1 - cos(\pi\theta_{w,i}/\theta_{fc,i}))^2$ for $\theta_{fc,i} \geq \theta_{w,i}$ and $\alpha_i^\theta = 1$ for $\theta_{fc,i} < \theta_{w,i}$. When the ground is frozen, sublimation is set to zero (i.e. $f = 0$), which in most real-world cases is not a limitation, as a snow cover builds up for which sublimation according to scheme 1 (see above) can occur.

4. In subsurface classes in which soil moisture is governed by Richards equation, water can flow upwards to compensate
for evaporative losses and all evaporated water is hence drawn from the uppermost grid cell. Similar to scheme 3, $f$ is set to $f = 0.25(1 - cos(\pi\theta_{w,1}/\theta_{fc,1}))^2$, while $r_e = r_a$ is assumed and the specific humidity is set to

$$q_1 = \exp\left(\frac{\psi_1 g}{R_{\text{wv}}(T_1 + T_{\text{mfw}})}\right) q_{\text{sat}}(T_1) \tag{12}$$

(Philip, 1957), where $R_{\text{wv}}$ is the gas constant for water vapor [J kg$^{-1}$ K$^{-1}$], $g$ is the gravitational acceleration [m s$^{-2}$], and $\psi_1$ [m] the matric potential of the uppermost grid cell (Sect. 2.2.3). Note that this scheme can only represent
evaporation, and should be combined with a dedicated vegetation module, such as the one demonstrated in Stuenzi et al. (2021a), to also represent transpiration.





*Lower boundary*: The lower boundary condition is applied to the lowermost class in the stratigraphy. In all classes described in this study, a user-defined constant heat flux $F_{\text{lb}}$ [$\text{W m}^{-2}$] is added to the lowermost grid cell which corresponds to the geothermal heat flux, $Q_{\text{geo}}$ [$\text{W m}^{-2}$], for sufficiently deep model domains. Although not yet implemented, it is possible to create classes with temperature boundary conditions similar to the upper boundary (see above).

### 2.2.3 Subsurface heat transfer and temperature calculation

*Heat conduction*: Depending on the selected stratigraphy class, CryoGrid considers heat conduction as well as heat advection as the dominant modes of heat transport in the subsurface. Thus, the change of enthalpy $e$ (see Sect. 2.2.1) is given by the continuity equation

$$\frac{\partial e}{\partial t} = -\frac{\partial j_{hc}}{\partial z} - \frac{\partial j_{\text{hw}}}{\partial z}, \tag{13}$$

with $z$ [m] the vertical coordinate, $j_{hc}$ the flux due to heat conduction and $j_{hw}$ the flux due to heat advected by water. Heat conduction is calculated from Fourier's law as

$$j_{hc} = -K_h \frac{\partial T(e)}{\partial z}, \tag{14}$$

with $K_h$ [$\text{W m}^{-1}\text{K}^{-1}$] the thermal conductivity and temperature $T$ as a function of $e$ (see "Soil freezing characteristics" below). The flux from heat advection with water flow is calculated as

$$j_{hw} = c_w T(e) j_w, \tag{15}$$

with $c_w$ [$\text{J K}^{-1}\text{m}^{-3}$] the volumetric heat capacity of liquid water and $j_w$ [$\text{m s}^{-1}$] the water flux, which consists of a term for vertical advection and a term for evapotranspiration (Sect. 2.2.4). For the thermal conductivity $K_h$, different parameterizations in terms of the volumetric contents of water, ice, mineral, organics and air can be selected by the user. For soil material, the parameterization by Cosenza et al. (2003) implemented in CryoGrid 2 and 3 is available, as well as the parameterization used in the Community Land Model (CLM) 4.5 (Oleson et al., 2013). While the former treats all soil constituents equal and thus functions for pure material, e.g. ice or rock, the latter is strictly focused on soils with reasonable porosity values. It first computes thermal conductivities for dry and saturated soil which is then weighted with the Kersten number to yield the final conductivity value (Johansen, 1973). For snow, the thermal conductivity is computed as a function of snow density, with two parameterizations available, namely the exponential relationship described in Yen (1981) and the quadratic relationship from Sturm et al. (1997).

*Soil freezing characteristics*: The soil freezing characteristics is a constitutive relationship between soil temperature and the unfrozen water content. In the CryoGrid community model, we generalize this concept to derive soil temperature $T$ and water content $\theta_w$ from enthalpy $e$ and water plus ice content $\theta_{wi}$ in the diagnostic step (Sect. 2.1.2). Depending on the subsurface class, either the "free water" freezing characteristic, or the soil freezing characteristic described in Painter and Karra (2014) is





implemented. In the free water case, all phase change of water occurs at $0\,^\circ\mathrm{C}$, i.e.

$$
T = \begin{cases}
e/c & \text{for } e \geq 0 \\
0 & \text{for } -L_{\mathrm{sl}}^{\mathrm{vol}}\theta_{wi} \leq e < 0 \\
(e + L_{\mathrm{sl}}^{\mathrm{vol}}\theta_{wi})/c & \text{for } e < -L_{\mathrm{sl}}^{\mathrm{vol}}\theta_{wi}
\end{cases}
\tag{16}
$$

and

$$
\theta_w = \begin{cases}
\theta_{wi} & \text{for } e \geq 0 \\
\theta_{wi}(1 + e/(L_{\mathrm{sl}}^{\mathrm{vol}}\theta_{wi})) & \text{for } -L_{\mathrm{sl}}^{\mathrm{vol}}\theta_{wi} \leq e < 0 \\
0 & \text{for } e < -L_{\mathrm{sl}}^{\mathrm{vol}}\theta_{wi}
\end{cases}
\tag{17}
$$

Here, $L_{sl}^{\mathrm{vol}}$ denotes the volumetric latent heat of freezing $[\mathrm{J\,m^{-3}}]$ and $\theta_{wi}$ the sum of the volumetric water and ice contents.

For the soil freezing characteristic by Painter and Karra (2014), the free water case functions are employed for $e \geq 0$. For $e < 0$, unique function $\theta_w(T, \theta_{wi})$ relating liquid water content to $T$ and $\theta_{wi}$ exist which we use to calculate $e(T, \theta_{wi})$ according to Eq. (1). For $e(T, \theta_{wi})$, lookup tables are compiled which allow to efficiently evaluate the inverse function $T(e, \theta_{wi})$ (and $\theta_w(e, \theta_{wi})$, combining $T(e, \theta_{wi})$ and $\theta_w(T, \theta_{wi})$) in each time step, thus computing the diagnostic variables temperature and water from the prognostic variable enthalpy. $\theta_w(T, \theta_{wi})$ is calculated using the matric potential $\psi$ [m] which also governs water flow in subsurface classes based on Richards equation (see Sect. 2.2.4). First, the matric potential in unfrozen state, $\psi_0$, is evaluated with the van Genuchten-Mualem model (Van Genuchten, 1980) as

$$
\psi_0 = -1/\alpha \left( (\theta_{wi}/\phi)^{-1/m} - 1 \right)^{1/n},
\tag{18}
$$

with $\alpha$ $[\mathrm{m^{-1}}]$ and $n$ [-] soil type specific parameters, $m = 1 - 1/n$ and assuming no residual water. While $\psi = \psi_0$ for unfrozen soil ($T \geq 0\,^\circ\mathrm{C}$), the matric potential for freezing soil ($T < 0\,^\circ\mathrm{C}$) is calculated as

$$
\psi = \psi_0 + \beta \frac{L_{\mathrm{sl}}^{\mathrm{vol}}}{g\rho_w} \frac{T - T_{\mathrm{mfw}}}{T_{\mathrm{mfw}}}.
\tag{19}
$$

Here, $T$ is in unit Kelvin, $g$ $[\mathrm{m\,s^{-2}}]$ is the gravitational acceleration, $\rho_w$ water density $[\mathrm{kg\,m^{-3}}]$, and $\beta$ the ratio of ice-liquid to liquid-air surface tensions for non-colloidal soil, set to 2.2 as suggested in Painter and Karra (2014). The water content is finally calculated as:

$$
\theta_w = \phi \left( 1 + (-\alpha\psi)^n \right)^{-m}.
\tag{20}
$$

The values of $\alpha$ and $n$ are determined by the soil type, and only a limited number of different soil types is possible within a stratigraphy class due to the need for lookup tables which are specific for combinations of $\alpha$ and $n$. Currently, four soil types (sand, silt, clay, peat) are implemented to provide users with a convenient interface, but it is possible to change the $\alpha$ and $n$ values associated with each of them, so that also other soil types can be realized.





### 2.2.4 Water balance

In the CryoGrid community model, three schemes to compute the time dynamics of soil water contents are available, namely 1. no flow (i.e. constant water plus ice contents), 2. a "bucket" scheme with only downward vertical water flow driven by gravity, and 3. vertical water flow governed by Richards equation. For schemes 2 and 3, the hydrological boundary conditions at the top of the soil column, such as rainfall input $P_l$, snow melt and evapotranspiration (related to the latent heat fluxes, Sect. 2.2.2) drive the time dynamics of the soil water content. Therefore, these only work in conjunction with the surface energy balance as upper boundary condition (Sect. 2.2), while scheme 1 can be applied for both temperature and surface energy balance boundary conditions, as in CryoGrid 2 (Westermann et al., 2013) and the initial version of CryoGrid 3 (as in Westermann et al., 2016). Within the soil domain, the time dynamics of the sum of water and ice contents is governed by water fluxes $j_w$ according to the continuity equation

$$\frac{\partial \theta_{wi}}{\partial t} = -\frac{\partial j_w}{\partial z}. \tag{21}$$

The three water balance schemes differ in their representation of $j_w$, which generally consists of vertical water fluxes $j_w^v$ and fluxes due to evapotranspiration $j_w^{ET}$ (or evaporation $j_w^E$ and transpiration $j_w^T$).

1. For the *no flow* scheme, $j_w = 0$, i.e. the sum of water and ice contents is fixed for each grid cell (and thus only determined by the initialization) and not affected by rainfall, snowmelt and evaporation. The no flow scheme therefore needs to be combined with either the temperature boundary condition or the surface energy balance with scheme 2 for evaporation (Sect. 2.2.2).

2. In the *bucket scheme*, the water in a grid cell is either immobile and bound to the soil matrix, or it flows downwards driven by gravity. The threshold between the two regimes is the user-defined field capacity, $\theta_{fc}$. In the unsaturated domain, the vertical water flux is hence given by

$$j_w^v = \begin{cases} -K_w & \text{for } \theta_w > \theta_{fc} \\ 0 & \text{for } \theta_w \leq \theta_{fc} \end{cases} \tag{22}$$

with $K_w$ the hydraulic conductivity $[\mathrm{ms^{-1}}]$. It is not the goal of the bucket scheme to reproduce the exact time dynamics of infiltration event and the hydraulic conductivity is broadly set to $K_w = K_{w,\mathrm{sat}}\theta_w/\phi$, with $K_{w,sat}$ $[\mathrm{ms^{-1}}]$ the saturated hydraulic conductivity specified by the user. This in particular prevents or slows infiltration in frozen-ice-saturated ground in spring, when the water content is low. No vertical water flux occurs in the saturated domain, unless water losses due to evapotranspiration must be compensated, i.e. $j_w^v = \max(-K_w, j_w^{ET})$. In subsurface classes representing soil, the bucket scheme is combined with scheme 3 for evapotranspiration (Sect. 2.2.2). The water flux due to evapotranspiration from grid cell i is calculated from the latent heat flux $Q_e$, as

$$j_{w,i}^{ET} = -\frac{f_i}{\sum_i f_i} \frac{Q_e}{L_{lg}\rho_w}, \tag{23}$$





with $f_i$ calculated from the water availability coefficients $\alpha_i^\theta$ (see scheme 3 in Sect. 2.2.2 for the other variables) as

$$f_i = f_{\text{tr}}\alpha_i^\theta \Delta z_i e^{-d_i/d_{\text{tr}}} + f_{\text{ev}}\alpha_i^\theta \Delta z_i e^{-d_i/d_{\text{ev}}}. \tag{24}$$

In essence, this ensures that a water flux corresponding to the weight of a grid cell in the calculation of the latent heat flux is extracted, taking water availability and exponential damping with depth into account. Note that CryoGrid 3 features a different bucket scheme, which does not treat soil water as a prognostic variable, but redistributes water in the bucket after each timestep (Nitzbon et al., 2019). As an example, a rain event leads to an instant increase of the water level in CryoGrid 3, while an infiltration front penetrating downwards with time is simulated by the subsurface classes available in the CryoGrid community model.

3. For water flow governed by *Richards equation* (Richards, 1931), movement of water in unsaturated soils through vertical gradients of the matric and gravitational potentials is accounted for, in addition to gravity-driven flow in the saturated domain. In this water balance representation in the CryoGrid community model, evaporation is drawn from the uppermost grid cell, i.e. $j_{w,1}^{\text{E}} = Q_e/(L_{\text{lg}}\rho_w)$. If transpiration is considered (e.g by the canopy scheme described in Stuenzi et al., 2021a), a grid cell weighting similar to Eq. (23) is used to compute the transpiration flux from each cell. The vertical water fluxes are calculated according to Richards equation

$$j_w^v = -K_w\left(\frac{\partial\psi}{\partial z} + 1\right), \tag{25}$$

using the matric potential $\psi$ which also accounts for soil freezing (see Sect. 2.2.3). For the hydraulic conductivity, we use the classic formulation by Van Genuchten (1980),

$$K_w = K_{\text{w,sat}}\, I_{\text{ice}}(\theta_w/\phi)^{0.5}\left(1 - \left(1 - (\theta_w/\phi)^{n/(n+1)}\right)^{(n-1)/n}\right)^2, \tag{26}$$

with an additional ice impedance factor $I_{\text{ice}} = 10^{-\Omega\theta_i/\theta_{wi}}$ (defined as in Hansson et al., 2004) to account for the blocking of water-filled pores by ice (see Sect. 3.1.3). In some subsurface classes, the permeability of the subsurface material, $k_w$ [m$^2$], needs to be specified instead of the saturated hydraulic conductivity, which is calculated according to $K_{\text{w,sat}} = k_w/(\eta_w\rho_w g)$. Here, $\eta_w$ is the temperature-dependent dynamic viscosity of water derived as $\eta_w = A\exp(B/T + CT + DT^2)$, with $T$ in unit Kelvin and coefficients $A, B, C, D$ as defined in Reid et al. (1987).

## 2.2.5 Excess ground ice

The CryoGrid community model comprises a subsurface class to simulate melting of excess ground ice which results in the subsidence of the ground surface. It is based on the bucket water scheme with freezing characteristic and surface energy balance (Sects. 2.2.2 to 2.2.4) and in most aspects similar to the CryoGrid 3 excess scheme (Westermann et al., 2016) based on Lee et al. (2014). However, freezing and melting of excess water/ice is treated differently than pore water/ice that is contained in the sediment matrix. While the pore water/ice freezes and melts according to the soil freezing characteristic, the excess water/ice portion is always treated as free water, i.e. it undergoes phase change at $T = 0\,°\text{C}$ (see Sect. 2.2.3). Two additional



state variables $\theta_{\chi i}$ and $\theta_{\chi w}$ denote the volumetric fractions of excess ice and water, so that $\theta_m + \theta_o + \theta_w + \theta_i + \theta_{\chi i} + \theta_{\chi w} + \theta_a = 1$.

The initial excess ice content is specified by the user and the excess ice fraction in a grid cell is unchanged (i.e. neither increases or decreases) as long as its temperature is below $0\,°C$.

Once excess ice melts, the excess water is mobilized and transported by the hydrology scheme, with an additional vertical water flux term $j^v_{\chi w} = K_w$ directed upwards. This excess water is first routed between the excess water variables of adjacent grid cells, with grid cell thickness changing accordingly (i.e. shrinking for net outflow, expanding for net inflow). If excess

water exists in an unsaturated grid cell (i.e. it contains a non-zero air content), water is moved from the excess water to the water phase, reducing the air content and leading to the grid cell thickness to shrink. In the uppermost grid cell, the excess water variable can be regarded as water pooling up above the surface, either due to melted excess ice routed upwards, or from rainfall and melted snow. This excess water can either evaporate, be routed away laterally (Sect. 2.3), or evolve into a pond/lake represented by a water body class (see Sect. 2.2.6). The two latter depend on the user-defined model setup which specifies what

happens when the excess water in the first grid cell exceeds a threshold depth.

In the user interface, the amount of excess ice in a subsurface grid cell is specified as a fraction ($\chi$) relative to the amount of soil without excess ice, i.e. $\chi = 1$ corresponds to a cell consisting of $50\,\%$ soil and $50\,\%$ excess ice.

### 2.2.6   Snow cover

Three stratigraphy classes representing snow are currently available in the CryoGrid community model, which all employ heat

conduction and the free water freezing characteristic (Sec. 2.2.3) to calculate snow temperatures. New snow is added to the first grid cell which is split into two cells when the "ice depth" (defined as $\theta_i \Delta z$, equivalent to snow water equivalent for dry snow) exceeds 1.5 times a user-defined target value (with the lower cell containing the target ice depth and the upper cell the remaining part). When the ice depth in a grid cell decreases below half the target ice depth, it is merged with the grid cell below. Meltwater becomes mobile when the volumetric water content exceeds a user-defined field capacity (provided as fraction of the

porosity of the ice matrix, $1 - \theta_i$) and can flow downwards, but also laterally if a corresponding lateral interaction class (Sect. 2.3) is selected. The dynamic interaction of snow classes with other stratigraphy classes, including their creation upon snowfall and removal when all snow has melted is described in Sect. 2.1.3. Of the three snow classes, class (a) can be combined with subsurface classes with temperature boundary condition and classes (b) and (c) with subsurface classes with surface energy balance (Sect. 2.2.2).

a) *Constant snow density, temperature boundary condition and degree-day based melt model*: In this snow class, new snow is added with a user-defined constant density. Temperature calculations inside the snow pack rely on a temperature boundary condition, heat conduction and the free water freeze curve. For snow melt, a degree-day based melt model is employed, using a melt factor calculated from latitude and day of year (as in Obu et al., 2019). The product of day length and solar culmination angle (i.e. the highest sun angle above the horizon at a given day of year) is used as a

measure of snow melt activity, which scales the degree day melt factor between confining values of $0.002\,\mathrm{m}(°C\,\mathrm{day})^{-1}$





and $0.012\,\mathrm{m}(°\mathrm{C\,day})^{-1}$ water equivalent. The snowmelt is assigned to the uppermost grid cell from where meltwater is removed once it exceeds the pore space, without infltrating into the snow pack.

b) *Constant snow density, surface energy balance and snow hydrology (bucket scheme)*: This snow class largely follows the snow parameterization of the CryoGrid 3 model, as described in detail in Westermann et al. (2016). As for (a), snowfall is added with prescribed density, but the surface energy balance is used as upper boundary condition with a transient albedo that decreases from a maximum value for fresh snow to a minimum value for old snow, with decrease rates depending on whether the snow is dry or wet. Furthermore, shortwave radiation penetrates into the snowpack following de Beer's law with a defined extinction coefficient, and sublimation/resublimation derived from the latent heat flux is extracted/added to the uppermost grid cell. The snow hydrology follows the bucket scheme (Sect. 2.2.4), with water from both rainfall and snowmelt percolating downwards when the water content exceeds the field capacity. In the simple snow cover module, refreezing of meltwater is the only process that can alter the density of a snow layer.

c) *Snow microphysics, surface energy balance and snow hydrology (bucket scheme)*: Introduced within CryoGrid 3 by Zweigel et al. (2021), this snow class is based on the Crocus snow scheme (Vionnet et al., 2012), including transient snow grain property and density evolution. The defining equations and parameterizations are largely identical to the ones described in Vionnet et al. (2012), so we only provide a brief description, concentrating on aspects treated differently. As for the previous class, the energy transfer at the upper boundary is prescribed according to the surface energy balance, but relies on spectrally resolved calculation of albedo and shortwave penetration and absorption in each grid cell. Snowfall is added with properties (density, grain size, dendricity and sphericity) derived from air temperature and wind speed, and the snow within each grid cell evolves and metamorphoses based on internal temperature gradient, water content, and the mass of overlying snow layers. The class also accounts for the impact of wind drift on snow grain properties and density, which in particular leads to compaction and density increase of the uppermost snow layers. If the uppermost snow grid cell features liquid water, both evaporation and sublimation are calculated and their fractions linearly interpolated between $\theta_w = 0$ (all sublimation) and $\theta_w = \theta_{\mathrm{evap}}^{\mathrm{snow}}$ (all evaporation). The threshold $\theta_w = \theta_{\mathrm{evap}}^{\mathrm{snow}}$ is set to twice the field capacity, but this should be revisited in future studies. For evaporation, the corresponding amount of water is extracted, while the same happens for the ice phase for sublimation. The original Crocus setup described in Vionnet et al. (2012) is associated with a variety of model parameters, some of which Royer et al. (2021) suggested to revise to better reproduce snowpack characteristics in the Arctic. These are in particular related to the wind speed dependence of the new snow density and the compaction dynamics due to wind drift, as well as parameterization of the snow thermal conductivity. In the snow class, it is possible to choose between the parameter sets for the original (according to Vionnet et al., 2012) and the "Arctic Crocus" (according to Royer et al., 2021), or to independently adjust the parameters in question. With this, the performance of either scheme within the CryoGrid community model can be evaluated against observations (see Sect. 3.2).





While the most process-rich scheme c) is generally expected to deliver a superior performance, it is also more sensitive to biases in the model forcing, especially the wind speed which strongly impacts snow density. In some cases, it is therefore be

preferable to employ the simpler schemes a) or b), especially if field measurements constraining the snow density are available.

### 2.2.7  Water bodies

In water bodies, heat transfer from the surface is strongly different between the ice-free and ice-covered seasons. The key characteristics of this seasonal asymmetry were conceptualized in CryoGrid 3 (see Westermann et al., 2016) for the highly relevant case of shallow water bodies. During the ice-free-season, the water column is assumed to be fully mixed due to wind

action, while a stable, temperature-driven stratification forms below an ice cover, with water at $0\,°C$ at the ice interface less dense than the warmer water at deeper layers. Within this water column, heat conduction is the main pathway of energy transfer with the relatively small thermal conductivity of water ($K_{h,w} = 0.57\,\mathrm{W\,m^{-1}\,K^{-1}}$) severely restricting energy losses of the ground below the water column. The CryoGrid community model provides a water body class based on the CryoGrid 3 model physics. In fact, the two seasonal regimes are implemented as two separate stratigraphy classes, which mutually create and

destroy each other upon defined conditions (see below). In the "ice-free class", the entire water column is simply represented by a single grid cell which assumes well-mixed conditions. The surface energy balance is applied at the upper boundary, and short-wave radiation penetrates into the water column with a bulk (i.e. not spectrally resolved) absorption coefficient. Both rain and snowfall are added to this grid cell with their respective enthalpy $e$, leading to both a change in temperature and in the grid cell thickness (and thus the water level). When the enthalpy reaches $e = 0$ (which is ensured by the time-stepping scheme,

Sect. 2.2.9), an ice cover forms and the ice-free class is exchanged by the ice-covered class. In this process, all state variables are split to the pre-defined model grid, so that the surface energy balance is now applied to the uppermost grid cell which can subsequently freeze according to the free water freezing characteristics. While energy transfer in the water and ice column is by means of heat conduction, grid cells are reordered after each timestep, with fully frozen cells (i.e. the ice cover) always on top and the unfrozen cells arranged by their temperature-dependent densities (according to Kell, 1975). When all ice has

melted, i.e. $e \geq 0$ for all grid cells, the grid cells are merged into a single grid cell and the ice-free class resumes. Note that the FLake water body scheme presented in CryoGrid 3 (Langer et al., 2016), is not yet available in the CryoGrid community model, but will be implemented in the future as an additional stratigraphy class.

### 2.2.8  Glaciers

The CryoGrid community model contains a glacier class, which consists of layers of pure ice (using the free water freezing

characteristic) with a user-defined constant ice thickness. Energy transfer is governed by heat conduction, with the surface energy balance as upper boundary condition. The scheme is usually coupled to snow schemes b) or c) (section 2.2.5), which allows the buildup of a seasonal snow layer for simulations of the ablation area, or if run over longer periods the buildup of a firn layer for simulations of the accumulation zone. The densification scheme in Crocus has been implemented into several models for simulations of snow and firn densification on glaciers (e.g. Cullather et al., 2016; Langen et al., 2017; Verjans et al.,

2019), which have been successfully applied for e.g. the Greenland ice sheet, Antarctica, and Icelandic glaciers (e.g. Agosta





et al., 2019; Fettweis et al., 2017; Schmidt et al., 2017). Similar applications are conceivable in the CryoGrid community model, with the glacier class representing ice and the Crocus-based snow class firn and the seasonal snow cover.

Water cannot infiltrate into the ice, so any water which does not refreeze during the time step will run off instantaneously, if there is no snow on the glacier surface. Likewise, if snow is present, liquid water in the snow class will build up above the
glacier ice, where it can eventually refreeze or run off, depending on the selected lateral interaction class.

If additional ice is added to the surface grid cell, either from refreezing of rain water or deposition, mass is advected downwards to ensure constant water equivalent water thickness. Similarly, if mass is removed from the module by runoff, evaporation, or sublimation, mass is advected up, with the lowest model layer receiving additional mass from an infinite ice reservoir below the model domain. This reservoir is assumed to have the same temperature as the lowest model layer. The
movement of mass within the model column is accompanied by a vertical transfer of sensible heat. A similar approach has previously been used in other glacier models which do not account for glacier flow (e.g. Langen et al., 2017), in order to prevent glacier areas with highly negative mass balance from disappearing during spin-up. However, the glacier class can also be employed without this option and coupled to a subsurface class representing subglacial sediments. In this case, the glacier can completely melt away, exposing the ground below, which for example offers the possibility to study glacier-permafrost
interactions (Sect. 4.5).

### 2.2.9 Numerical implementation

For prognostic variables like enthalpy and water content, we use the integral form of the respective continuity equations (Eqs. 13 for heat and 21 for water). For a scalar, volume-normalized quantity $s$ (e.g. $e$ for enthalpy and $\theta_w$ for water contents), the time change within a volume $V$ is explained by fluxes $\vec{j}_s$ across surface $\Omega_V$ of $V$, with normal vector $\vec{n}$:

$$\int_V \frac{\partial s}{\partial t} dV + \oint_{\Omega_V} \vec{j}_s \cdot \vec{n} d\Omega_V = 0. \tag{27}$$

The numerical implementation is based on finite differences with grid cells (index $i$ increasing downwards, vertical thickness $\Delta z$ [m] and area $A$ [m$^2$]) within which $s$ is considered constant. Furthermore, for one-dimensional simulations, only vertical fluxes through the upper and lower boundary of the grid cell have to be considered, so that the continuity equation for grid cell $i$ simplifies to

$$\frac{\partial}{\partial t} S_i := \frac{\partial}{\partial t}(A\Delta z_i s_i) = -j_s^{i-1\leftrightarrow i} + j_s^{i\leftrightarrow i+1}, \tag{28}$$

i.e. the time derivative of the bulk quantity $S_i$ is simply obtained from the fluxes $j_s^{i-1\leftrightarrow i}$ (defined positive when directed upwards) across the interfaces between grid cells $i-1/i$ and $i/i+1$. For the numerical implementation in the CryoGrid community model, it is therefore of practical advantage to use the extensive bulk quantities as state variables and not the volume-normalized quantities (which are employed as direct model state variables in e.g. CryoGrid 2/3) for which the defining differential equa-
tions in the previous sections are provided. In the CryoGrid community model, each stratigraphy class covers an explicit area $A$ [m$^2$], and the the model state variables for mineral, organic, water and ice contents become volumes: $\phi_m = A\Delta z\theta_m$ [m$^3$];





$\phi_o = A\Delta z\theta_o$ [m$^3$]; $\phi_w = A\Delta z\theta_w$ [m$^3$]; $\phi_i = A\Delta z\theta_i$ [m$^3$]. Likewise, the bulk value for the enthalpy for each grid cell is used, $E = A\Delta z e$ [J]. For time integration of Eq. (28), we use a simple first-order forward Euler scheme as in CryoGrid 3 (Westermann et al., 2016), i.e.

$$S_i(t + \Delta t) = S_i(t) + \Delta t \left(-j_s^{i-1\leftrightarrow i} + j_s^{i\leftrightarrow i+1}\right). \tag{29}$$

Stability and accuracy are guaranteed by selecting small enough time steps with conditions specifically designed for each state variable, the particular requirements of the model physics of each stratigraphy class, and the typical orders of magnitude (and timescales of change) of model forcing and parameters. For enthalpy, for example, a maximum change of volume-normalized enthalpy $e$ between grid cells is defined by the user, and time steps $\Delta t$ are adjusted to not exceed this value for any grid cell.

Therefore, small timesteps are generally required if large fluxes occur, slowing down computation. In a similar manner, changes in soil water content between timesteps can be limited, while it is also possible to prevent "overfilling" of a grid cell (so that the water content exceeds the pore space) by limiting the time step accordingly. Another example is the water body class (Sect. 2.2.7) for which the time step calculation guarantees, that the condition $e = 0$ (which triggers the switch between ice-free and ice-covered water body classes, Sect. 2.2.7) is exactly met. In addition, a maximum timestep can be defined to satisfy the CFL

condition (Courant et al., 1928) for the parameters (e.g. thermal conductivities and heat capacities) and grid cell sizes of the simulation setup. The fluxes of both heat and water have the general form

$$j_s = -\kappa_s(z)\frac{\partial \sigma(z)}{\partial z}. \tag{30}$$

Conductivities $\kappa_s$ are defined for individual grid cells, while $j_s$ in the finite difference scheme is expressed in terms of fluxes between grid cells $i-1$ and $i$, $j_s^{i-1\leftrightarrow i}$. For his reason, the effective conductivities governing the flux between grid cells are

calculated as series of the resistances $1/\kappa_{s,i-1}$ and $1/\kappa_{s,i}$ using half of the grid cell thicknesses $\Delta z$:

$$j_s^{i-1\leftrightarrow i} = \frac{2\kappa_{s,i-1}\kappa_{s,i}}{\kappa_{s,i-1}\Delta z_i + \kappa_{s,i}\Delta z_{i-1}}(\sigma_i - \sigma_{i-1}). \tag{31}$$

In the upper- and lowermost grid cells of the CryoGrid stratigraphy (which are in different stratigraphy classes if the stratigraphy consists of more than one class), the fluxes derived at the upper and lower boundaries (e.g. $F_{\text{ub}}$ and $F_{\text{lb}}$ for heat) are added. To connect adjacent stratigraphy classes, corresponding fluxes $j_s^{N_u\leftrightarrow 1_l}$ are calculated by the interaction class (Sect. 2.1.2), with

grid cell $N_u$ being the lowermost grid cell of the upper stratigraphy class, while grid cell $1_l$ is the uppermost grid cell of the lower stratigraphy class. $j_s^{N_u\leftrightarrow 1_l}$ depends on the state variables and model physics of both classes involved. It can be of the same form as Eq. (31), but also $j_s^{N_u\leftrightarrow 1_l} = 0$, for example the water flux between a stratigraphy class with and one without water balance.

## 2.3 Lateral interactions with an external environment

With CryoGrid 3, several studies were presented that simulate lateral exchange of energy and matter (heat, water or snow), either with external reservoirs for single-tile simulations (Martin et al., 2019; Langer et al., 2016), or between different CryoGrid stratigraphies for three-dimensional multi-tile configurations (Martin et al., 2021; Nitzbon et al., 2019, 2020, 2021; Zweigel





et al., 2021). In the CryoGrid community model, we extend these possibilities by providing a standardized interface to implement a variety of lateral interactions, which are compatible with the CryoGrid stratigraphy consisting of a stack of classes

(Fig. 4). This functionality is accomplished by two further types of classes, "lateral classes" and "lateral interaction classes", both of which are selected in the TILE_1D_STANDARD class. The choice of the lateral class determines whether the Cryo-Grid stratigraphy interacts with external (and static) reservoirs ("LATERAL_1D"), or whether several CryoGrid stratigraphies interact with each other ("LATERAL_3D"), which corresponds to the "laterally coupled tiling" demonstrated in CryoGrid 3. In this study, we focus on interactions with external reservoirs, while laterally coupled tiling with LATERAL_3D will be de-

scribed in a separate study in the future. Other than for the time integration in the vertical Cryogrid stratigraphy, lateral fluxes are added/subtracted after a fixed, user-determined interaction timestep (which is a parameter in the lateral class), so they are not part of the time integration scheme of the regular stratigraphy. This is largely due to the computation-related requirements of laterally coupled tiling (which requires parallel computing, see Nitzbon et al., 2019), but constant timesteps $\Delta t^{lat}$ are also employed for one-dimensional simulations with external reservoirs to ensure consistency. In general, lateral fluxes exchanged

with external reservoirs should be small compared to the corresponding vertical fluxes within the CryoGrid stratigraphies, which means that interaction timesteps can be significantly longer than the typical timesteps required for the vertical integration. The lateral interaction timestep must be selected by the user, seeking for a balance between runtime and model accuracy and stability. Typical lateral timesteps for the applications presented in Sect. 3.2 are between half an hour to six hours.

The lateral class sets up the environment for the lateral interaction classes which represent the actual model physics of the

lateral interactions with external reservoirs. Lateral interaction classes can be combined with each other, with the lateral class calling them one after the other in the order provided by the user (as a list in TILE_1D_standard).

### 2.3.1 Lateral coupling to heat reservoir

The CryoGrid stratigraphy can be laterally coupled to an external heat reservoir, which can be used to mimic the thermal impact of infrastructure, water bodies (similar to Langer et al., 2016) or more general adjacent areas with strongly different ground

thermal regime, e.g. at the edge of a permafrost-underlain peat plateau (Martin et al., 2021). The heat reservoir is characterized by a user-defined, constant temperature $T^{\text{lat}}$ and lateral distance $d^{\text{lat}}$ [m] from the CryoGrid stratigraphy. Furthermore, a lower and an upper elevation for the heat reservoir must be provided, which makes it possible to confine the effect of the heat reservoir to a part of the stratigraphy. Furthermore, several heat reservoirs with different temperatures and upper/lower elevations can be combined, which achieves a similar effect as the coupling to a temperature stratigraphy (Langer et al., 2016). If a grid cell $i$ is

located between the lower and upper bounds of the heat reservoir, the lateral heat flux is calculated as

$$j_{\text{hc,i}}^{\text{lat}} = -K_{\text{h,i}} \frac{T_i - T^{\text{lat}}}{d^{\text{lat}}}, \tag{32}$$

with $K_{\text{h,i}}$ the thermal conductivity of grid cell $i$ and $T_i$ its temperature. The change in bulk enthalpy in grid cell $i$ over time step $\Delta t^{\text{lat}}$ is given by

$$\Delta E_i = \Delta t^{\text{lat}} \Delta z_i l_c^{\text{lat}} j_{\text{hc,i}}^{\text{lat}}, \tag{33}$$





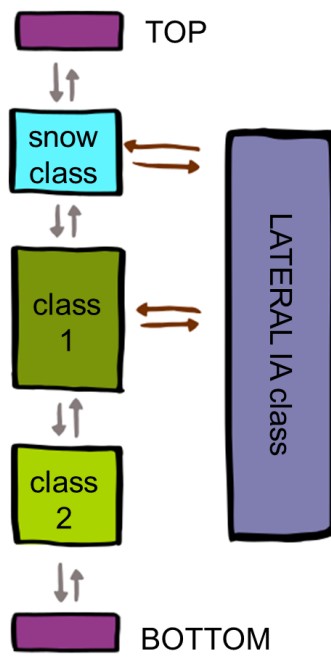

**Figure 4.** Schematic representation of CryoGrid stratigraphy (for the example of a fully developed snow cover) interacting with a lateral interaction (IA) class (see text). Note that it is specific to each stratigraphy class if and how it is modified by the lateral interaction class. In this example, stratigraphy class 2 is unaffected, while the snow class and class 1 are modified by the lateral interaction class.

with $l_c^{lat}$ [m] the lateral contact length and $l_c^{\text{lat}}\Delta z_i$ [m$^2$] the cross section through which the lateral heat flux occurs.

### 2.3.2  Lateral water transport

*Surface water removal*: When snow melts or rain falls on a saturated soil column, water will either pool up on the surface or is lost as surface runoff. The exact way to treat this surface water depends on the particular stratigraphy class, with most classes removing surface water (but storing it in a state variable), while for example the excess ice class (Sect. 2.2.5) can explicitly
represent a surface water pool. The CryoGrid community model provides a lateral interaction class that constantly removes any surface water and stores the accumulated surface runoff, irrespective of the way surface water is treated in the uppermost stratigraphy class. This in particular makes it possible to generate an unbroken time series of surface runoff which comprises both snow classes (i.e. snowmelt runoff) and subsurface classes during the snow-free season.

*Overland flow*: For the excess ice class (Sect. 2.2.5), standing surface water can be represented as excess water in the
uppermost grid cell, with water depth given as $d_w = \theta_{\chi w}\Delta z$ [m]. Instead of removing surface water like with the previous interaction class, surface water can also be removed by overland flow governed by the Gauckler-Manning equation (Gauckler,





1867; Manning et al., 1890) as

$$j_{\text{w,1}}^{\text{lat}} = -G\,d_w^{2/3}(\delta^{\text{lat}})^{1/2}, \tag{34}$$

with $G$ [m$^{1/3}$] the Gauckler-Manning coefficient, and $\delta^{\text{lat}}$ [-] the local gradient which drives the surface flow. The change in
bulk water content in grid cell 1 over time step $\Delta t^{\text{lat}}$ is given by

$$\Delta\Theta_{w,1} = \Delta t^{\text{lat}} d_w l_c^{lat} j_{\text{w,1}}^{\text{lat}}, \tag{35}$$

with $l_c^{lat} d_w$ [m$^2$] the cross section through which the lateral water flux occurs ($l_c^{\text{lat}}$ is the lateral contact length, see above).
Accordingly, the change in bulk enthalpy due to water advection is given by

$$\Delta E_1 = \Delta t^{\text{lat}} d_w l_c^{lat} j_{\text{w,1}}^{\text{lat}} c_w T_1. \tag{36}$$

*Seepage face*: For stratigraphy classes with water balance (Sect. 2.2.4, schemes 2 and 3), a seepage face lateral boundary
condition is implemented as a lateral interaction class, leading to drainage in the saturated domain of the soil column. The
lateral interaction class first determines the elevation of the water table, $z_{\text{wt}}$, and then removes a lateral water flux

$$j_{w,i}^{\text{lat}} = -K_{w,i}\frac{z_{\text{wt}} - z_i}{d^{\text{lat}}} \tag{37}$$

for grid cells $i$ with elevations $z_i < z_{\text{wt}}$. Here, $K_{w,i}$ is the hydraulic conductivity and $d_{lat}$ the lateral distance to the seepage face
which determines the strength of the drainage. In unsaturated grid cells, no outflow occurs. Note that the water table elevation
is tracked across stratigraphy classes, e.g. when the water table is located in a water body class, it also governs the outflow from
the subsurface class below. The seepage face always leads to outflow and it is possible to define the upper and lower elevations
for the domain through which outflow occurs. The change in bulk water content in grid cell $i$ over time step $\Delta t^{\text{lat}}$ is given by

$$\Delta\Theta_{w,i} = \Delta t^{\text{lat}} \Delta z_i l_c^{\text{lat}} j_{w,i}^{\text{lat}} \tag{38}$$

with $l_c^{\text{lat}} \Delta z_i$ [m$^2$] the cross section through which the lateral water flux occurs. Accordingly, the change in bulk enthalpy due
to water advection is given by

$$\Delta E_i = \Delta t^{\text{lat}} \Delta z_i l_c^{\text{lat}} j_{w,i}^{\text{lat}} c_w T_i. \tag{39}$$

*Water reservoir*: Similar to seepage flow, hydrological coupling to an external water reservoir is possible, located at elevation
$z^{lat}$ and lateral distance $d^{\text{lat}}$. The lateral water flux is calculated as

$$j_{w,i}^{\text{lat}} = -K_{w,i}\frac{z_i - z^{\text{lat}}}{d^{\text{lat}}}, \tag{40}$$

with parameters as described above. As for seepage flow, water flow is restricted to the saturated zone, but inflow can occur if
$z_i < z^{\text{lat}}$. This lateral interaction class can therefore be used to keep the soil water table at a certain level, with in- and outflow
depending on rainfall and evapotranspiration. While outflowing water has the temperature of the respective grid cell, inflowing





water can optionally be assigned a (constant) reservoir temperature $T^{lat}$, which is taken into account in terms of heat advection
through water. In this case, the change in bulk enthalpy is given by

$$\Delta E_i = \Delta t^{\text{lat}} \Delta z_i l_c^{\text{lat}} j_{w,i}^{\text{lat}} c_w T^{\text{lat}}, \tag{41}$$

while the change in bulk water content $\Theta_{w,i}$ is the same as for the seepage face (see Eq. 38).

## 3   Results

### 3.1   Benchmarking of selected model components

#### 3.1.1   Step-change in upper boundary temperature

To document the basic numerical performance of the model framework, we model the temperature response of an infinite
half-space to a step change in temperature at the upper boundary. An analytical solution to this problem is available (Carslaw
and Jaeger, 1959) and is given by:

$$T(x,t) = T_{\text{init}} + (T_{\text{ub}} - T_{\text{init}}) \operatorname{erfc}\left(\sqrt{\frac{c\,x^2}{4K_h t}}\right), \tag{42}$$

where $T(x,t)$ [°C] is the temperature at time $t$ [s] and depth $x$ [m] below the surface, $T_{\text{init}}$ [°C] and $T_{\text{ub}}$ [°C] are the initial
domain temperature and temperature applied at the upper boundary, respectively. $c$ and $K_h$ are the heat capacity [$\mathrm{J\,m^{-3}\,K^{-1}}$]
and thermal conductivity [$\mathrm{W\,m^{-1}\,K^{-1}}$] of the medium, and erfc is the complementary error function. The formulation is valid
for a homogenous material with no variation in thermal properties in space and time, and without phase change.

In CryoGrid we simulate the response using a stratigraphy class with zero heat flux at the lower boundary and temperature
boundary condition (Sect. 2.2.2) at the upper boundary. A $100\,\mathrm{m}$ deep model domain is selected, discretized with a spacing of
$0.1\,\mathrm{m}$, and initialized with a constant temperature of $T_{\text{init}} = 1\,°\mathrm{C}$. At $t = 0$, an upper boundary temperature of $10\,°\mathrm{C}$ is applied,
and the simulation is allowed to run for five years.

Fig. 5 shows the model and analytical responses for selected time steps. Using a maximum change in internal energy per time
step of $50\,\mathrm{kJ\,m^{-3}}$ (which is the default CryoGrid setting used in all following simulations), the maximum absolute difference
between the analytical and numerical simulations is less than $0.003\,°\mathrm{C}$. The maximum error is reduced to less than $0.0003\,°\mathrm{C}$
by decreasing the maximum energy step to $10\,\mathrm{kJ\,m^{-3}}$.

#### 3.1.2   Stefan problem

In 1889, the Slovene physicist Josef Stefan published an analytical solution to the one-sided freezing problem and applied it
to sea-ice formation in the Arctic Ocean (Stefan, 1891). The solution was reformulated and adapted for modeling the thawing
of frozen soils by Nixon and McRoberts (1973). To obtain an analytical solution, the following simplifying assumptions were
introduced: 1) the heat capacity of the medium is neglected, 2) the temperature in the medium below the freezing front does

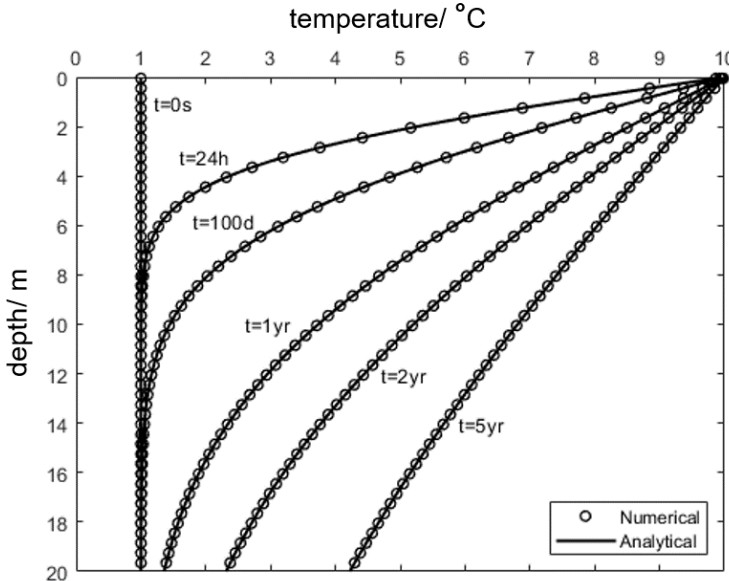

**Figure 5.** Comparison of analytical and numerical simulations of a step change in temperature at the upper boundary of a homogeneous half-space. Results are shown the initial condition (t = 0 s) and for 24 h, 100 days, and 1,2, and 5 years after the step change. Numerical results are plotted for every 4th grid cell.

not affect the rate of thaw, 3) the temperature distribution in the thawed medium is linear, and 4) a constant surface temperature is applied. With these assumptions, the thaw depth as a function of time is given by (Nixon and McRoberts, 1973):

$$d_{\text{thaw}} = \sqrt{\frac{2K_h T_{\text{ub}}}{L_{\text{sl}}^{\text{vol}} \theta_{wi}}} \sqrt{t}, \tag{43}$$

where $d_{\text{thaw}}$ [m] is the thaw depth at time $t$ [s], $K_h$ [W m$^{-1}$ K$^{-1}$] the thermal conductivity of the medium (considered constant with temperature), $T_{\text{ub}}$ [°C] the upper boundary temperature, $L_{\text{sl}}^{\text{vol}}$ [J m$^{-3}$] the volumetric latent heat of fusion and $\theta_{wi}$ [-] the initial volumetric water (for freezing) or ice (for thawing) contents.

   This analytical solution to the problem of top-down thawing of frozen soils, which is often referred to as the Stefan solution, has been used here to benchmark the CryoGrid model for a situation with phase change. We used a stratigraphy class with

a free water soil freezing characteristic (Sect. 2.2.3), temperature boundary condition (Sect. 2.2.2) and zero heat flux at the lower boundary. The domain was initialized with an initial temperature of $-0.02$ °C which is a good approximation to satisfy assumption 2. The volumetric ice content was chosen to be 0.3, and the mineral content 0.7. The upper boundary temperature was fixed at 1 °C, to minimize the effect of the heat capacity of the medium (to align with assumption 3). The domain was discretized with a grid spacing of 0.01 m from the surface to 3 m depth, and 0.1 m from 3 m depth to the lower boundary at

100 m depth. The model was run with a constant surface temperature for 5 years, and the numerical and analytical results are compared in Fig. 6.



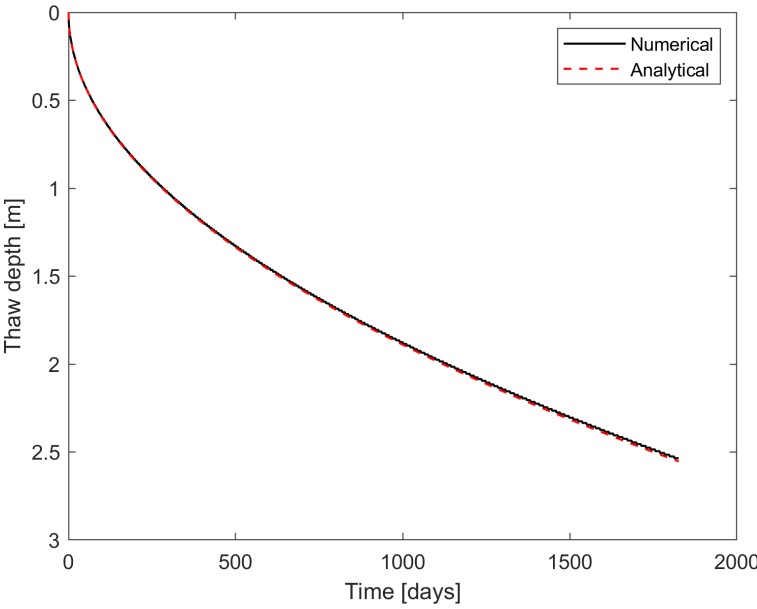

**Figure 6.** Comparison of analytical and numerical simulations of one-sided thawing of a frozen soil column. The soil temperature is initalized at $-0.02\,°\mathrm{C}$ and an upper boundary temperature of $1\,°\mathrm{C}$ is applied for a period of 5 years. The shown domain is discretized with a node spacing of $0.01\,\mathrm{m}$.

The numerical and analytical solutions are nearly identical, with the numerically derived thaw depths slightly shallower than the ones derived from the analytical solution. The fact that the numerical formulation accounts for the heat capacity of the medium and thus storage of sensible heat, while the analytical solution does not, accounts for the very small offset observed
towards the end of the period.

### 3.1.3 Mizoguchi (1990) experiment on cryosuction

The Mizoguchi (1990) experiment is a classic model benchmark for cryosuction, i.e. the redistribution of soil water during soil freezing. A number of unfrozen, $0.2\,\mathrm{m}$ cm long soil columns with constant soil water content are exposed to one-sided freezing, while the other side is kept insulated. After 12, 24 and 50 hours, the combined content of water and ice is determined
gravimetrically for one of the columns, so that the water redistribution over time can be followed. The experimental conditions including suggestions for model setup are presented in detail in Hansson et al. (2004). We modified the upper and lower boundary conditions of the subsurface class with water balance according to Richards equation (Sect. 2.2.4) accordingly, setting the heat flux at the lower boundary ($0.2\,\mathrm{m}$) to zero. The heat flux at the upper boundary is set proportional to the temperature difference between the first grid cell and the coolant (at $-6\,°\mathrm{C}$). As in Hansson et al. (2004), we tried two settings
for the proportionality coefficient (i.e. the convective heat transfer coefficient), one constant at $28\,\mathrm{W\,m^{-2}\,K^{-1}}$ (denoted 'linear heat transfer scenario') and one decreasing nonlinearly from 40 (above $0\,°\mathrm{C}$) to $10\,\mathrm{W\,m^{-2}\,K^{-1}}$ (below $-4\,°\mathrm{C}$) as a function





of the first grid cell temperature squared (denoted 'non-linear heat transfer scenario'). The soil porosity was set to $0.535$ with zero organic content and we use an initial water content of $0.345$ to match the initial state (0 h, see Fig. 7) depicted in Hansson et al. (2004). Similar to Painter (2011), the soil permeability is set to $3.25 \times 10^{-13}\,\mathrm{m}^2$, which corresponds to a saturated

hydrological conductivity of $3.2 \times 10^{-6}\,\mathrm{m\,s}^{-1}$ (Hansson et al., 2004) for a water viscosity at room temperature. For the silty soil employed in the experiment, we used the van Genuchten parameters $\alpha = 1.11\mathrm{m}^{-1}$ and $n = 1.48$ (Hansson et al., 2004). For the thermal conductivity, we use the parametrization of Cosenza et al. (2003), with a thermal conductivity of the mineral fraction of $2\,\mathrm{W\,m}^{-1}\,\mathrm{K}$. This roughly reproduces the measured frozen and thawed thermal conductivities for volumetric water contents of $0.4$ and $0.3$, while thermal conductivities for lower water/ ice contents are overestimated (note that such low water

contents are not reached in the simulation).

Fig. 7 shows the best fit to the measurements which is achieved with the nonlinear heat transfer scenario. Except from small deviations near the freeze front for the 12 and 24 hour states, CryoGrid manages to reproduce the measurements very well. Using the linear heat transfer scenario (Suppl. 3, Fig. S1) leads to only small changes in the results, with almost equally good fit. Likewise, changing the thermal conductivity parametrization to the one used in CLM4.5 modified the results only

marginally (not shown). Most importantly, the parametrization of the hydrological conductivity had a pronounced influence on the results, especially the choice of the additional ice impedance factor $I_{\mathrm{ice}}$ (Sect. 2.2.4). The best fit (Fig. 7) is achieved for $\Omega = 5$, i.e. the hydrological conductivity is decreased by factor of $10^{-5}$ when water contents approach zero. With the default factor of $\Omega = 7$ (Dall'Amico et al., 2011), water redistribution is considerably weaker, resulting in a notably worse fit to the measurements (Suppl. 3, Fig. S2). While this should be investigated in more detail, we have set $\Omega = 5$ in all simulations with

Richards equation in this study.

### 3.1.4 Accelerated spin-up to reach steady-state temperature profile

In most real-world examples of thermal simulations, the initial temperature profile is not known and must be estimated by a model spin-up, i.e. by running the model for a certain time period (denoted the spin-up period) until the simulated temperature profile becomes independent of the initial profile (i.e. it is only determined by the model forcing applied at the upper and lower

boundaries). While this state is usually reached within a few years in the uppermost meters, it takes much longer for deeper layers. For climate change simulations at centennial timescale with a typical model domain depth of $100\,\mathrm{m}$ (e.g. Westermann et al., 2016), a spin-up of several hundred years can be necessary, thus requiring significant additional computation time. However, reliable spin-up model forcing this long back in time is often not available, so that it is approximated by repeatedly looping a shorter period, often the first part of the regular model forcing. For this practically relevant case, the CryoGrid

community model offers the possibility to considerably accelerate model spin-up by estimating the equilibrium temperature profile for the spin-up period, broadly following the procedure outlined in Westermann et al. (2013, 2017) for the CryoGrid 2 model:

1. use the TTOP model (Sect. 2.2.2) to obtain a first estimate for the temperature at the top of the permafrost/bottom of the seasonally frozen layer, $T_{\mathrm{TOP,1}}$;



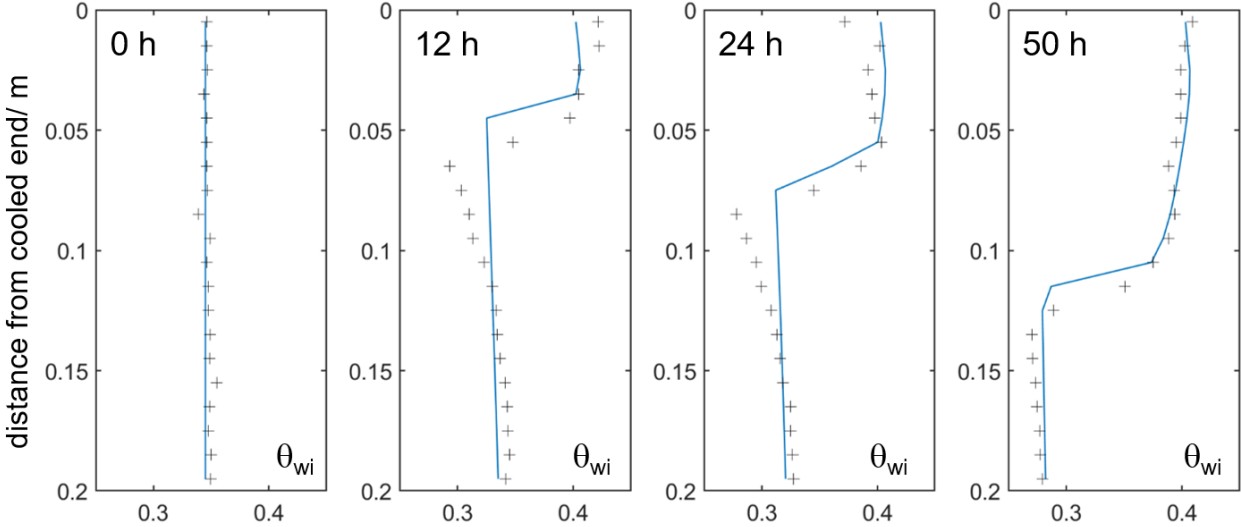

**Figure 7.** Simulated sum of volumetric water and ice content (blue lines) vs. measurements (crosses, digitized from Hansson et al., 2004) for the Mizoguchi (1990) experiment for 0, 12, 24 and 50 hours freezing time; nonlinear heat transfer scenario, ice impedance factor calculated with $\Omega = 5$.

2. estimate a plausible seasonal thaw/freeze depth, apply $T_{\mathrm{TOP,1}}$ above and compute a steady-state temperature profile below with the lower boundary heat flux and thermal conductivities computed for the respective temperatures;

3. run the model twice for the selected spin-up period and store freezing and thawing degree days (FDD and TDD) for each model grid cell for the second iteration;

4. determine the seasonal thaw/freeze depth and the temperature at the top of the permafrost/bottom of the seasonally frozen layer, $T_{\mathrm{TOP,2}}$, from FDD and TDD;

5. repeat step 2 with $T_{\mathrm{TOP,2}}$ and the seasonal thaw/freeze depth calculated in the previous step;

6. run the model once for the selected spin-up period, prior to starting the regular simulations.

As only steps 3 and 6 require significant computation, the total runtime of the accelerated spin-up approximately corresponds to three times the normal model runtime for the spin-up period. Fig. 8 displays an example for the accelerated spin-up procedure, using a period of ten years. For comparison, a classic spin-up is performed, starting with a constant initial temperature of $-8\,°\mathrm{C}$ throughout the entire profile (which is clearly too cold, considering the model forcing). While the classic spin-up takes 200 to 400 years (i.e. 20 to 40 iterations of the ten-year period) to reach a reasonable approximation of the targeted equilibrium state, the accelerated spin-up requires only 30 model years (three iterations of spin-up period) to reach a similar performance. Fig. 8 shows that the accelerated spin-up is not exact, with the temperature at 100 m depth about $0.1\,°\mathrm{C}$ warmer than the classic spin-up after 400 years which is still cooling slightly at this point. The reason for this is that the thermal conductivity in the uppermost



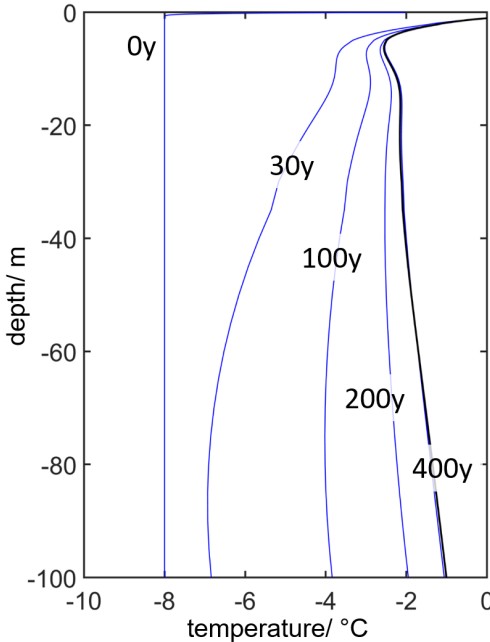

**Figure 8.** Comparison of accelerated and classic spin-up for a ten-year period 1980, July 31 to 1990, July 31; simulated temperature profile on July 31 for accelerated spin-up (black, achieved with 30 simulation years) and classic spin-up (blue) after 0, 30, 100, 200 and 400 simulation years. Setting: Svalbard forcing (see Sect. 3.2) and ground stratigraphy, Crocus-based snow class, 100 m deep model domain with geothermal heat flux of $50\,\mathrm{mW\,m^{-2}}$ at the lower boundary.

permafrost layers still fluctuate annually with temperatures, giving rise to a small additional thermal offset (Romanovsky and Osterkamp, 1995). However, this small deviation of the accelerated spin-up is not relevant for real-world applications, as the true temperature profile does not correspond to the equilibrium state for the spin-up period anyway, but is determined by the real climate conditions in the past. Therefore, the accelerated spin-up procedure in the CryoGrid community model provides a
compromise between computation time and initialization accuracy that is fully sufficient for many applications.

### 3.2 Example simulations for Svalbard

To demonstrate different configurations of modular CryoGrid stratigraphies, we perform simulations for the well studied Bayelva permafrost research site outside Ny-Ålesund, Svalbard (Boike et al., 2018). A soil and climate monitoring station established in 1998 provides records of active layer temperatures, meteorological variables and snow depth. The Bayelva site
is situated on top of a hill, within a short distance (ca. 1 km) to the Brøggerbreen glacier. The unglaciated coastal areas are underlain with continuous permafrost extending to depths of about 100 m and with active layer depths of 1-2 m. The landscape is characterized by patterned permafrost ground and sparse vegetation coverage. The soils on the hill range from silty loam to silty clay, while a coarser grained material (silty sands to gravel) in the surrounding area. The site includes a 9 m deep





borehole with hourly measurements of permafrost temperatures dating back to 2009, featuring a mean permafrost temperature
of $-2.8\,°C$. In the active layer and in the top of permafrost the annual mean temperatures increased by $1.8\,°C$ per decade for
the period 1998-2017 (Boike et al., 2018). The soil and air temperature trends show the largest temperature increase in winter
(Maturilli et al., 2015). At the Bayelva site, the snow cover build up starts in September/October and maximum snow depths
can reach up to $1.5\,m$. The timing of spring snowmelt can by vary several weeks and typically starts between May to June.
Data from the Bayelva site have been used for validation of various earth system modeling studies (Chadburn et al., 2017;
Ekici et al., 2014, 2015).

We prepare a common forcing time series for all simulations by downscaling surface fields from the ERA-5 reanalysis
(closest model grid cell), using the Bayelva measurements as reference. For each day of year, a linear regression is computed
between reanalysis data and available measurements within a window of 20 days before and after which is subsequently
employed to correct the reanalysis data. This procedure is in particular able to account for a potential seasonality of the bias
of the reanalysis data, but relies on a sufficient correlation between the two time series (Westermann et al., 2015, 2016). The
downscaling is applied to air temperature and incoming long-wave radiation for which the reanalysis is strongly cold-biased
during the summer period. The specific humidity is corrected accordingly for the change in temperature. Incoming short-wave
radiation and wind speed are used without downscaling, as their average values generally agree well with measurements, while
the correlation was lower due to their strong dependence on the timing of synoptic events which is poorly captured by the
reanalysis. Furthermore, precipitation is not corrected as there are no reliable measurements especially for snowfall. Instead,
we perform simulations for different multiplication factors for snowfall to evaluate the sensitivity to this important variable.
All simulations are performed for the period 1981 to 2018, using the accelerated spin-up procedure (Sect. 3.1.4) for the first
ten years to initialize the ground temperature profile.

In the following, we first establish a model baseline configuration that can reproduce measurements reasonably well (Sect.
3.2.1) and then change one model component at a time, while the others are kept as in the reference (Sect. 3.2.2). We emphasize
that only the model parameters of the reference simulation are optimized to match the measurements, while better-fitting
configurations likely exist for the other simulations.

### 3.2.1 Reference simulations

We use the model setup of Zweigel et al. (2021) for the wider area around the Bayelva station as a starting point to optimize
model settings specific for the location of the measurement station. A stratigraphy class with surface energy balance (Sect.
2.2.2), soil freezing characteristic (Sect. 2.2.3) and soil type silt, bucket water balance (Sect. 2.2.4) is used to represent sedi-
ments in the uppermost $5\,m$. Bedrock is assumed below, represented by a less process-rich stratigraphy class with free water
freezing characteristic (Sect. 2.2.3) and constant soil water/ice (Sect. 2.2.4). As the Bayelva station is located on a small hill,
lateral drainage with the lateral interaction class representing a seepage face (Sect. 2.3.2) is assumed. The seasonal snow is
represented by the Crocus-based snow class (described under c) in Sect. 2.2.6), with parameters for wind compaction, initial
snow density and short-wave albedo adapted to fit both measured ground surface temperatures, snow depths and pre-melt snow
densities of 300 to $350\,kg\,m^{-3}$ which is a rough average for measurements conducted in 2011 to 2015. Fig. 9 shows simulated



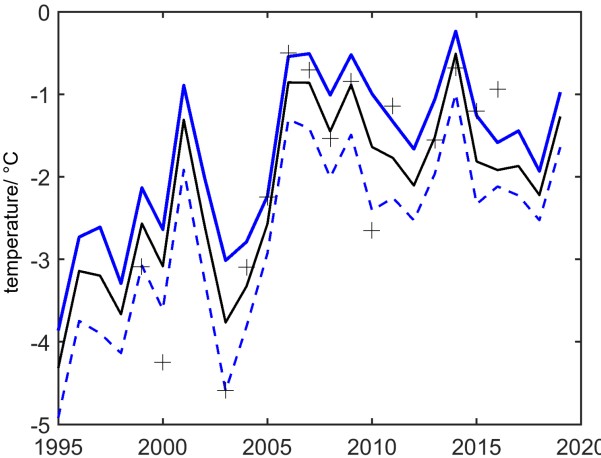

**Figure 9.** Simulated (lines) and measured (crosses) annual average temperatures at 1.3 m depth for the reference simulation (see text). Years with measurements available for less than 90 % of the time are not shown. Black line: 100 % snowfall; broken blue line: 90 % snowfall; solid blue line: 110 % snowfall (used as comparison to other model configurations, Sect. 3.2.2).

annual average temperatures at 1.3 m depth for snowfall multiplications factors of 0.9, 1.0 and 1.1. For the majority of the years, the measurements fall within this range which indicates that at least the general magnitude of snowfall is captured well

by the model forcing. Figs. 10 and 11 display the full time series of ground temperatures near the ground surface and at 1.3 m depth for the measurements and the simulations with snowfall multiplication factor of 1.1. The simulations can describe the seasonality of measured temperatures well for most of the years, but winter temperatures are considerably too warm for some years, e.g. 2000/2001, 2002/2003 and 2009/2010. In these years, however, the measured snow depths at the Bayelva station are lower than in the simulations for most of the snow season (Fig. 12) which is the likely reason for the overestimated ground

temperatures. In general, simulated snow depths increase smoothly throughout the winter season in the simulations, while the measurements bear evidence of individual snowfall events which lead to strong increases in snow depth.

### 3.2.2  Simulations with different CryoGrid stratigraphies

We use the warmest reference simulation with 110 % snowfall as baseline for which simulated ground temperatures are close to the thaw threshold in some years. In all model configurations displayed in this section, snowfall is hence increased to 110 %

as well. We then exchange one model aspect at a time, leaving all others unchanged so that differences in simulation results can be attributed to the model aspect in question. This is either accomplished by selecting a different stratigraphy class, or by changing model parameters. It is important to note that a poor performance in reproducing measurements can not necessarily be attributed to the investigated model aspect, as only the reference simulation is optimized for all model parts together. We also emphasize that the results of the model comparison are specific to the particular conditions for the Bayelva site and can not



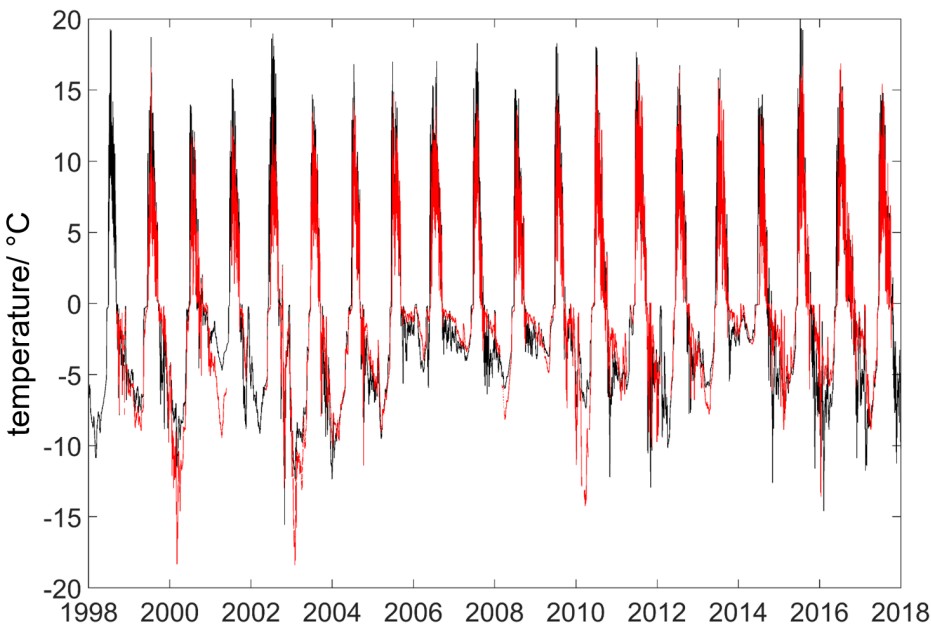

**Figure 10.** Simulated (black) and measured (red) ground temperatures at $4\,\mathrm{cm}$ depth, Bayelva soil and climate station, Ny-Ålesund, Svalbard (data from Boike et al., 2018). Reference simulations with $110\,\%$ snowfall, see text. Surface energy balance as upper boundary, subsurface module with soil freezing characteristic, and lateral drainage with seepage face, Crocus-based snow model with optimized parameters.

necessarily be generalized. Therefore, the main purpose of this comparison is to showcase the effect of various setups within the modular CryoGrid community model, and not to evaluate their performance at the Bayelva site in a strict sense.

*Subsurface representation and water balance*: Fig. 13 displays simulated ground temperatures for different soil freezing characteristic, from silt for the reference simulation to sand and finally free water freezing for which all phase change of water occurs at $0\,^{\circ}\mathrm{C}$. The differences in simulated ground temperatures are relatively small which suggests that the choice of the soil

freezing characteristic is not critical. At least in some situations, it is therefore possible to speed up simulations by using free water freezing instead of the computationally more demanding simulations with soil water freezing characteristic.

Secondly, we investigate the effect of different lateral drainage conditions when using the bucket water scheme, with the reference simulation representing well-drained ground conditions due to the applied seepage face draining (Fig. 14), while excess water pooling up at the surface is removed instantly. Secondly, the same configuration is simulated with the excess

water/ice class (Sect. 2.2.5), which allows water to pool up at the surface (and potentially refreeze), being removed laterally by overland flow (Sect. 2.3.2). Note that no excess ground ice is added to the ground itself, so that model physics and parameters are identical to the reference simulation, except for the representation of surface water. While the differences to the reference simulation are generally small, the simulated ground temperatures are both warmer and colder in individual years. The main reason for the differences is the representations of surface ice layers which regularly occur in fall and early winter in W



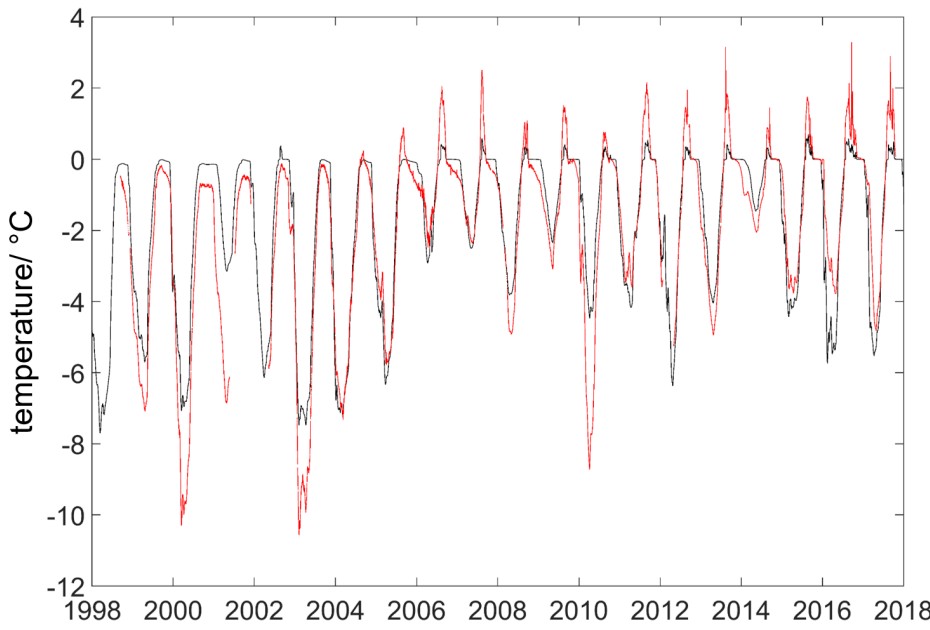

**Figure 11.** Simulated (black) and measured (red) snow depths at the Bayelva soil and climate station, Ny-Ålesund, Svalbard (data from Boike et al., 2018). Reference simulations with $110\,\%$ snowfall. See text.

Svalbard. If the ground is frozen, but snow-free during a rain event (and saturated with ice/water), the rain water instantly removed in the reference simulation, while it can freeze and form a surface ice layer in the excess ice class. This not only changes the freeze/thaw dynamics of the uppermost ground layer, but also delays the onset of thawing in spring, as the melting surface ice layer confines the temperature of the uppermost grid cell to $0\,°\mathrm{C}$. The impact on the ground thermal regime of these two effects strongly depends on the conditions of the individual year, which explains the both negative and positive differences

in the simulated mean annual ground temperatures. The third model configuration is once again based on the exact setup of the reference simulation, but without lateral in- or outflow of water, representing a classic, one-dimensional simulation without interactions with external reservoirs. In the fourth configuration (denoted "wet"), inflow of water from a reservoir at the height of the ground surface ensures permanently wet ground conditions, mimicking a wetland located in a depression. While there is little difference between the one-dimensional and the wet simulation case, they are both significantly warmer than the well-

drained reference simulation (Fig. 14) and a partly unfrozen zone above the permafrost develops in some of the years after 2006, corresponding to permafrost degradation. This can be explained by the higher soil water and thus latent heat content of the soil, which prevents complete freeze-back before an insulating winter snow cover forms. The small differences between the 1D and the wet simulation case is explained by the generally high summer precipitation at the Bayelva site which often exceeds evapotranspiration, thus keeping the ground wet. Furthermore, in the bucket scheme employed for this comparison, water is

not accessible to evapotranspiration below a certain depth which is determined by the parameters evaporation and transpiration



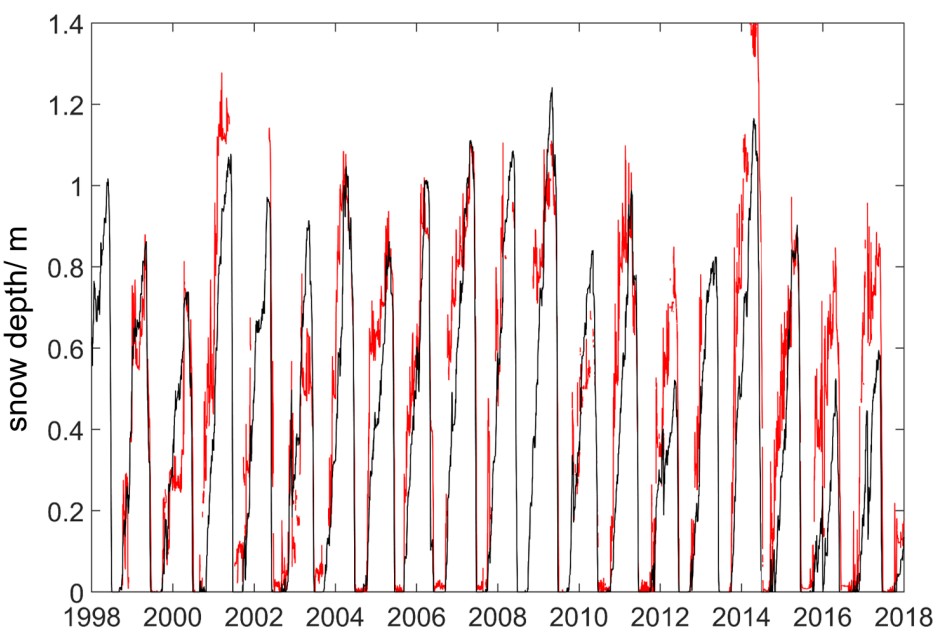

**Figure 12.** Simulated (black) vs. measured (red) ground temperatures at 133 cm depth, Bayelva soil and climate station, Ny-Ålesund, Svalbard (data from Boike et al., 2018). Reference simulations with 110 % snowfall, see text. Surface energy balance as upper boundary, subsurface module with soil freezing characteristic and lateral drainage with seepage face, Crocus-based snow model with optimized parameters.

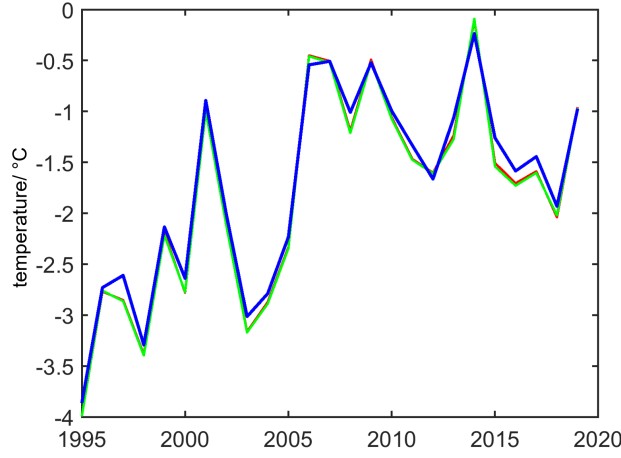

**Figure 13.** Simulated annual average temperatures at 1.3 m depth. Blue: reference simulation; green: as reference simulation, but with soil freezing characteristic for sand instead of silt; red: soil water freezing as free water.



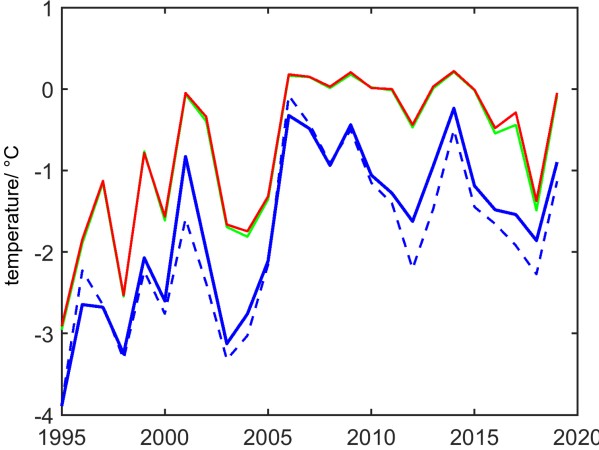

**Figure 14.** Simulated annual average temperatures at 1.3 m depth. Blue, solid line: reference simulation for drained soil conditions; blue, dashed line: as reference simulation, but representing surface water/ice (with excess ice module, Sect. 2.2.5) and lateral overland flow (Sect. 2.3.2); green: no drainage; red: wet soil with influx of water.

depth in the model. While this generally does not occur for a shallow active layer, a permanently saturated layer forms for the deep active layer in the Bayelva simulations, which requires a long time to freeze in fall and winter. When lateral drainage is applied, as in the reference simulations, the soil water content is held at field capacity instead, so that the freeze-back occurs faster, thus leading to colder ground temperatures and stable permafrost also after 2006.

The simulations not only deliver ground temperatures, as e.g. the CryoGrid 2 and 3 models, but it is also possible to evaluate the components of the water balance, consisting of precipitation $P$, evapotranspiration $ET$, runoff (both from surface and subsurface) $R$ and storage $S$ in the form of snow and soil water/ice, i.e. $P = ET + R + S$. In particular CryoGrid stratigraphies which allow for standing surface water and removal by overland flow are well suited for this task, as all lateral runoff is handled in a physically-based way (and not simply by removing excess water instantly). Fig. 15 displays the accumulated water balance

for three example years for this model configuration (dashed blue line in Fig. 14), showing the seasonality of the storage term corresponding to the winter snow cover, which is largely converted to runoff during and after snowmelt. At the same time, evapotranspiration is largest in July, corresponding to snow-free surface conditions and large amounts of incoming radiation. Smaller amounts of water equivalent are lost to sublimation during the snow-covered season which is well visible during the first winter 2009/2010. Note that there are also smaller changes in storage during the snow-free season which corresponds to

changes in soil water storage.

We further investigate the influence of the soil water balance on simulation results by considering different model representations (Fig. 16). The reference simulation with bucket water scheme is first compared to a simulation with constant soil water content set to the field capacity of the reference simulation, with evapotranspiration calculated independent of the soil water content by setting a constant model parameter for the "surface resistance against evapotranspiration" (as in the scheme

used for Westermann et al., 2016). The simulated soil temperatures are very similar to the reference simulation which reflects



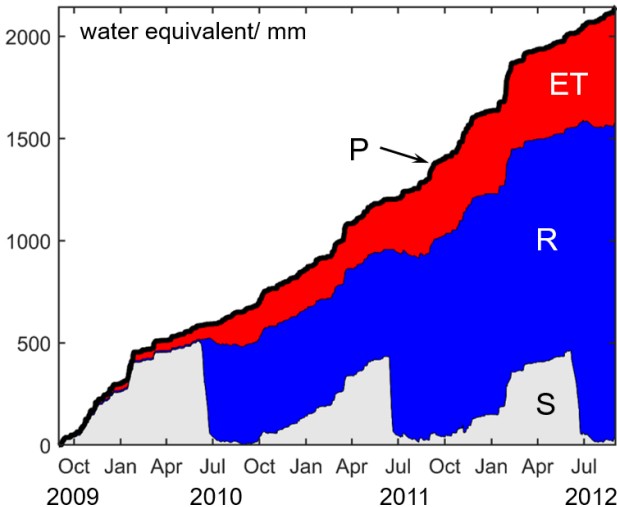

**Figure 15.** Simulations of accumulated water balance for simulations with surface water representation (i.e. excess ice class), lateral drainage by a seepage face and overland flow, as well as $110\,\%$ snowfall (dashed blue blue in Fig. 14) for three example years. P (thick black line): precipitation (sum of rain- and snowfall); ET: evapotranspiration; R: runoff (sum of surface and subsurface runoff); S: storage (soil and snow).

the overall similar water contents within the active layer. In addition, the reference simulation is compared to the hydrological scheme driven by Richards equation, including redistribution of soil water during the freezing process. While this results in slightly higher ground temperatures, they are still reasonably similar to the reference simulation (Fig. 16). Fig. 17 displays the simulated volumetric water contents for the three different model representations of the soil water balance, showing significant

differences during the unfrozen season. While the soil water content does naturally not change in the "constant water content" case (middle panel), approximately the upper half meter (controlled by the user-defined characteristic depths of evaporation and transpiration, Sect. 2.2.4) dries out as result of evpotranspiration for the bucket water scheme (left panel), while strong rainfall events periodically reset the water content to the field capacity. At the same time, water accumulates near the bottom of the active layer, which is removed laterally by the applied seepage face module. Finally, the entire active layer remains

unsaturated in the simulations with Richards equation, as water is redistributed upwards to compensate for evaporation (right panel) during most of the summer season. For this reason, no lateral subsurface runoff occurs (see Sect. 2.3.2), although the soil water content is overall higher than in the reference simulation, especially in fall. The simulated thaw depths are more or less equal for the bucket water and the Richards equation schemes, while they are slightly deeper for the scheme with constant soil water contents. We point out that the simulations with Richards equation yield a different assessment of permafrost stability in

the wider Bayelva area, despite the relatively similar ground temperatures: while permafrost in the reference simulation is only stable when lateral drainage occurs (Fig. 14), it is also stable for the one-dimensional case if the soil water balance is governed by Richards equation.



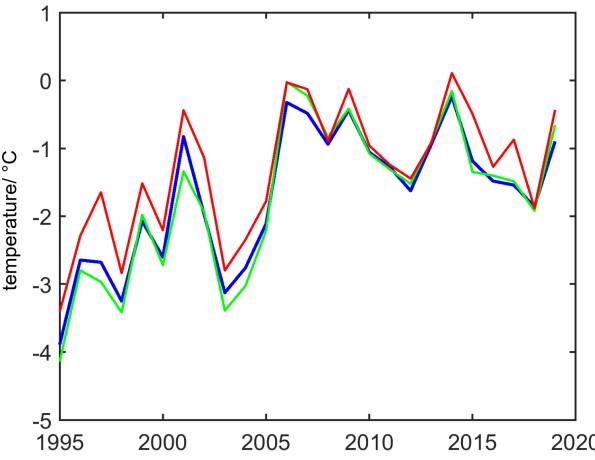

**Figure 16.** Simulated annual average temperatures at 1.3 m depth. Blue: reference simulation (bucket scheme for water, including lateral subsurface drainage); green: constant soil water plus ice content; red: Richards equation (no lateral subsurface drainage occurs).

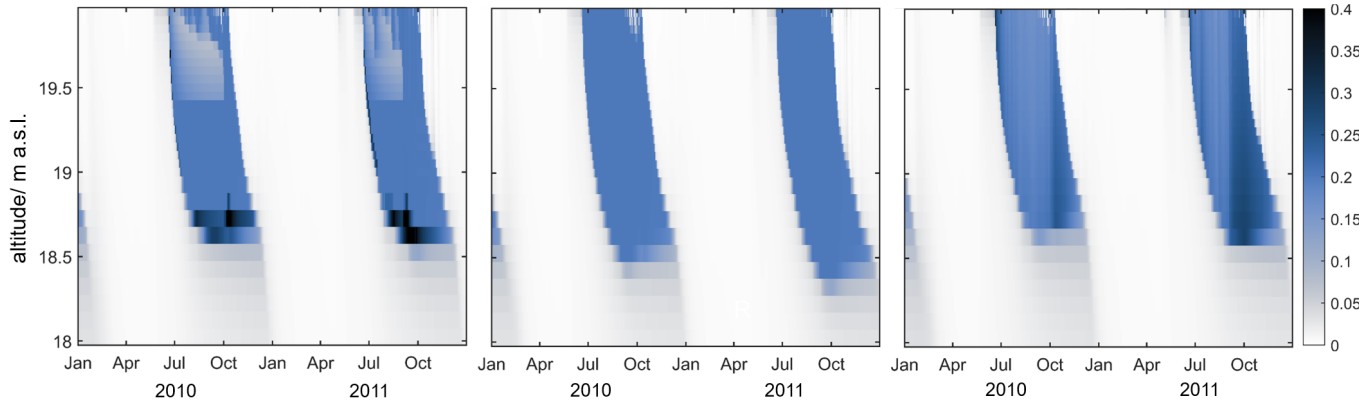

**Figure 17.** Simulated profiles of volumetric water contents for two example years for the three water balance schemes used in Fig. 16. Left: reference simulation (bucket scheme for water, including lateral drainage); middle: constant soil water plus ice content; right: Richards equation (no lateral subsurface drainage occurs).

An example for the effect of cryosuction in the Richards equation scheme is displayed in Fig. 18, which shows the sum of the volumetric water and ice contents, $\theta_{wi}$, for one example year. The accumulation of ground ice below and above the freeze

front is well visible in this year, which in particular leads to an ice-rich layer at the bottom of the active layer, which can modify thaw behaviour in the following years (compare to Schuh et al., 2017). Furthermore, cryosuction can replenish ground ice that has melted during summers with deep active layers and is lost to the surface due to evaporation (Fig. 17, right panel). The previous maximum of the active layer thickness is clearly visible in Fig. 18, and ice accumulation over several years with shallower active layer could replenish the lost ground ice at least partly. In the Bayelva simulations, this does not really happen

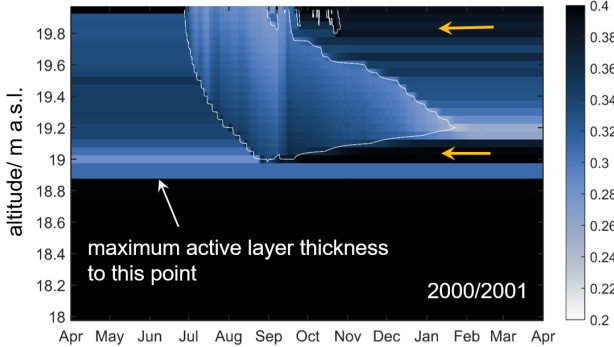

**Figure 18.** Simulated sum of volumetric water and ice contents using Richards equation for a one-year period in 2000/2001 (red in 16); white line: freeze front. Cryosuction upon soil freezing leads to ice accumulation above and below the freeze front, as marked by the two orange arrows. In this example period, the effect of cryosuction is particularly pronounced, while it is of smaller magnitude in other periods.

as the active layer is overall deepening during the simulation period, so that ground ice formed at the bottom of the active layer due to cryosuction eventually melts out.

*Snow cover representation*: Figure 19 displays the simulation results for different snow cover classes, in particular the class with snow microphysics representation (following the Crocus-based snow model) and the snow class with a constant snow density (as in the classic CryoGrid 2 and 3 models). Both snow classes feature a bucket scheme for water infiltration, including refreezing of meltwater percolating through the snow pack. While the parameters in the Crocus-based snow class are specifically adapted for the reference simulation, two of the parameter settings presented in literature produce too high ground temperatures and permafrost is not thermally stable. These are the original Crocus parameter set (Vionnet et al., 2012), which produces too low snow densities and thus too much insulation of the ground, and the "arctic" setting (Royer et al., 2021). This produces reasonable snow densities, but the employed parametrization for the snow thermal conductivity (after Sturm et al., 1997) yields significantly lower values than in the original Crocus configuration (using the parametrization after Yen, 1981). As a result, the simulated temperatures are relatively similar for the two settings, despite significant differences in the simulated snow densities and thus snow depths. Note that the reference simulation uses the parameter set of the "arctic" configuration, but the Yen-parametrization for the snow thermal conductivity. We point out that snow accumulation at the Bayelva site is subject to strong wind redistribution of snow which likely also changes over time, as the measurement site is located on a hill and thus exposed to snow ablation, but is also surrounded by a protective fence providing wind resistance and thus promoting deposition. The results must therefore be regarded with caution and should not be generalized unless backed up by simulations from other sites.

The simulations with constant snow densities are overall colder than in the reference, although overall low snow densities of 250 and $275\,\mathrm{kg\,m^{-3}}$ were used (compared to end-of-season measurements of 300 - $350\,\mathrm{kg\,m^{-3}}$), which represent a seasonal and depth-average for the entire snow pack. In fall and early winter this likely overestimates the snow density and thus thermal





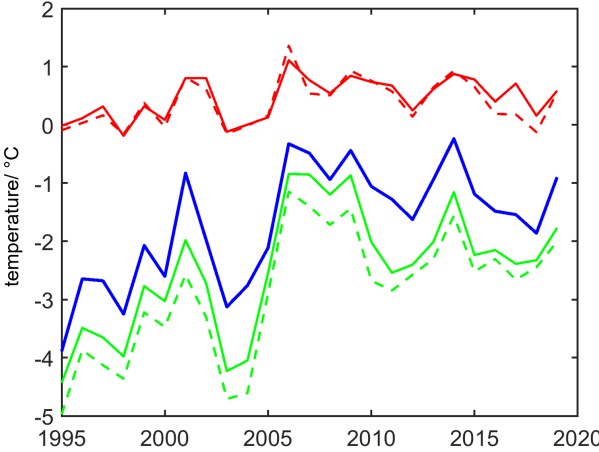

**Figure 19.** Simulated annual average temperatures at $1.3\,\mathrm{m}$ depth. Blue: reference simulation (Crocus-based snow scheme with adapted parameters); green solid: constant density snow scheme with initial density of $250\,\mathrm{kg\,m^{-3}}$; green dashed: constant density snow scheme with initial density of $275\,\mathrm{kg\,m^{-3}}$; red solid: Crocus-based snow scheme with original parameters from Vionnet et al. (2012); red dashed: Crocus-based snow scheme with "arctic" configuration (Royer et al., 2021).

conductivity of the fresh snow pack, so that the ground can refreeze faster. In the annual balance, this offsets the too strong insulation in the later parts of the winter, when the true snow densities are higher in the majority of the snow pack.

*Model type and upper boundary condition*: In the reference simulation, energy exchange at the upper boundary is driven by the surface energy balance, similar to the original CryoGrid 3 model. While most stratigraphy classes in the CryoGrid community model employ the surface energy balance, simpler approaches, such as a temperature boundary condition (as in CryoGrid 2) and the semi-empirical TTOP model (as in CryoGrid 1), are also available. Fig. 20 shows a comparison of different model types, with a CryoGrid 2 like model setup with air temperature as upper boundary condition for both the ground and the snow surface, and snow ablation derived from a degree-day melt model (Sect. 2.2.6). While the model does not account for the water balance and is generally a much less process-rich approach compared to the reference simulation, it delivers a similar representation of annual average temperatures for the Baylva site. Interestingly, even the simple TTOP approach can capture some of the warming trend observed in the last two decades, although the $n$-factors are not adapted to represent interannual differences of snow depths. However, as a stand-alone, this equilibrium approach should strictly speaking be only be applied to longer time periods (dashed green line in Fig. 20), as it can not account for the time lag of ground temperature changes due heat conduction and possibly ice melt. This problem can be partly overcome by using the TTOP model as upper boundary condition for a stratigraphy class simulating heat conduction (dotted green line in Fig. 20). In this example application, the TTOP model computes mean annual ground temperatures for annual time slices, which is used as temperature boundary condition for the heat conduction model (Sect. 2.2.2). While the active layer dynamics is not resolved, this hybrid approach can efficiently simulate the temperature dynamics in deeper layers (similar to Myhra et al., 2017) which is in particular interesting to speed



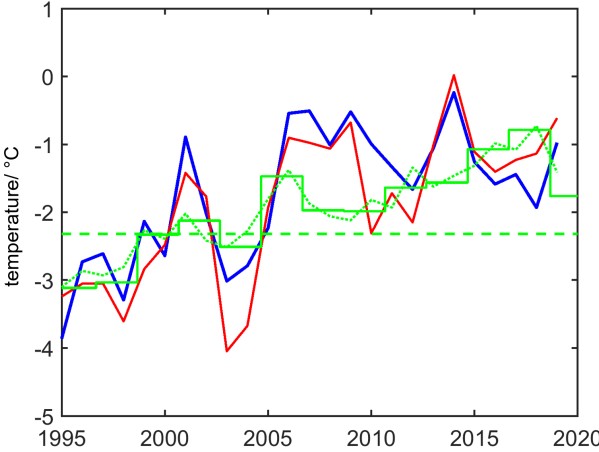

**Figure 20.** Simulated annual average temperatures at 1.3 m depth. Blue: reference simulation (upper boundary condition surface energy balance); red: temperature boundary condition model (air temperature as upper boundary condition, degree-day based snow melt model); green: TTOP model with time-constant n-factors $n_f = 0.5$, $n_t = 1$, $r_k = 0.8$ (dashed: entire period; solid: 2-year intervals; dotted: one-year interval as upper boundary condition for CryoGrid stratigraphy as in reference simulation).

up centennial and millennial-scale simulations. Although the TTOP model is a simplistic approach associated with significant

uncertainty, it is straight-forward to train and validate with more process-rich model setups in the CryoGrid community model.

*Glaciers and water bodies*: The CryoGrid community model provides stratigraphy classes for land surface types other than ground and snow, in particular glaciers and water bodies. Fig. 21 displays simulated ground temperatures for a glacier and a shallow (0.8m deep) water body when driven with the same forcing data as the reference simulation. The water body simulations clearly show that average ground temperatures are positive throughout the entire time which is due to the asymmetry of

heat transfer between summer and winter (Westermann et al., 2016). This suggests that there is a permafrost-free zone below water bodies of that depth around Ny-Ålesund, with the water body not freezing to the bottom in winter. Fig. 22 shows the simulated lake ice dynamics for two example years, with significantly thicker ice cover during a year with low overall low snow depths. However, even in this year, the water body does not freeze to the bottom, which explains the high ground temperatures and permafrost-free conditions. Conversely, annual average temperatures are significantly lower for the glacier than for the ref-

erence simulation, as the energy input during summer is consumed by ice melt, rather than warming of the subsurface material. Furthermore, the summer meltwater fully drains so that freezing temperatures in fall and early winter instantly penetrate in the subsurface, other than for soil freezing where the water in the active layer must first refreeze, before the ground below can cool out further. For the employed stratigraphy class, the annual ice melt is automatically compensated by mass from an infinite ice layer below the model domain. Fig. 23 shows the annual ice melt for a theoretical glacier located at the Bayelva site. The ice

melt follows the observed mass balance from nearby glaciers Austre Brøggerbreen and Midtre Lovènbreen well, which have an increasingly negative mass balance trend since the 1980's with the highest mass loss year in 2016.



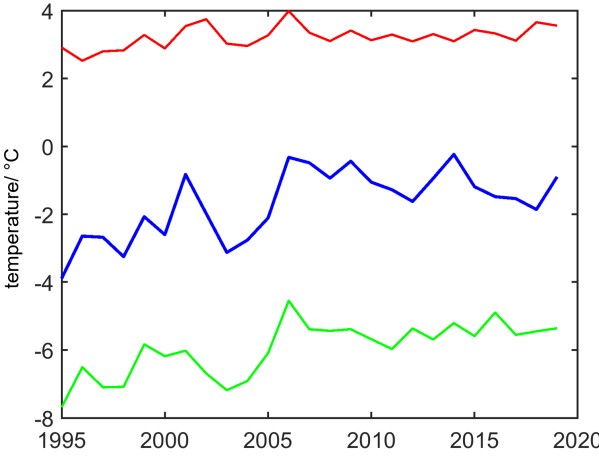

**Figure 21.** Simulated annual average temperatures at $1.3\,\mathrm{m}$ depth. Blue: reference simulation for soil material; red: 0.8 m deep water body overlaying soil material (as for the reference simulation); green: glacier.

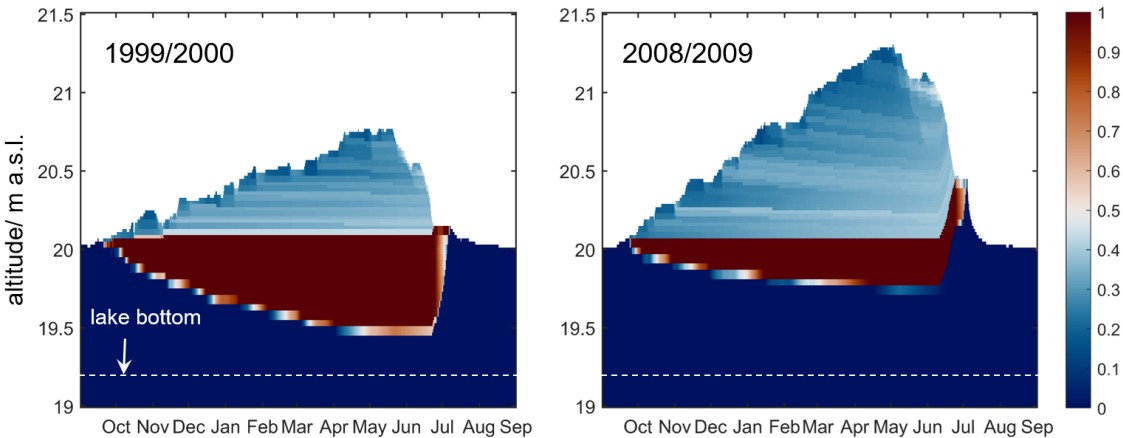

**Figure 22.** Simulated ice dynamics (volumetric ice content) for the water body simulation (red in Fig. 21) for a winter with low and high snow depths. As snow melt water is assumed to pass through the water body ice in the model, the ice rises in elevation during the melt period which is particularly evident in 2008/2009.

## 4 Discussion and Outlook

The modular concept of the CryoGrid community model was designed to unlock added value of model development conducted in several research projects with strongly different research focus, such as Nunataryuk, PermaRisk and ESA Permafrost_CCI. 990 Instead of building specialized model tools for a narrow set of applications, the community model aims for a library of different model parts that can be flexibly combined by users, while providing developers with an advanced starting point to design new



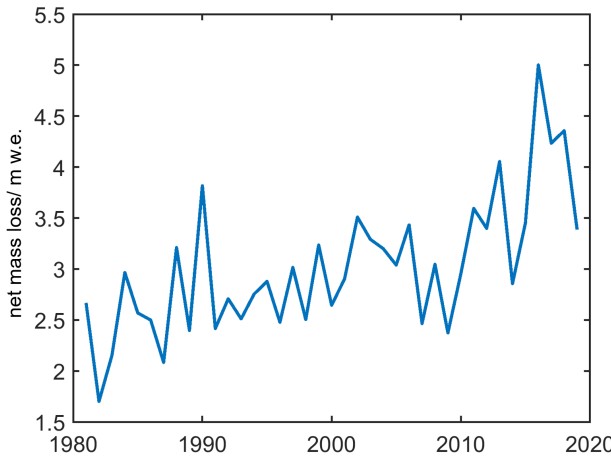

**Figure 23.** Simulated annual net ice melt (in meters of water equivalent) with the glacier module for the Bayelva forcing data set.

model functionalities. As such, the goal of the CryoGrid community model is to become a dynamic platform for continued model development, not a static model applied by its user community to solve the same classes of problems.

### 4.1 Differences to CryoGrid 1-3 model implementations

*CryoGrid 1*: The same TTOP equilbrium model that forms the core of CryoGrid 1 is implemented by a dedicated stratigraphy class in the CryoGrid community model. However, most CryoGrid 1 applications relied on additional parametrizations for the model parameters. In particular the winter $n$-factor (Eq. 3) has been calculated from snow depth (e.g. Gisnås et al., 2013), snowfall (Westermann et al., 2015), or snow depth and winter temperature (Obu et al., 2019). While it is straight-forward to add these parameterizations to the CryoGrid community model, they have all been developed and adapted for specific input

data sets. We therefore leave it to concrete future applications to complement the "raw" TTOP model currently implemented with additional parameterizations for its model parameters.

*CryoGrid 2*: The main features of CryoGrid 2 are a temperature boundary condition, a simple snow model with constant snow density and a numerical scheme optimized for computational efficiency. The first two features are retained in the corresponding stratigraphy classes of the community model, but the numerical scheme has been changed to the simple first-order forward

Euler scheme with adaptive timestep, which is not not as computationally efficient as previous CryoGrid 2 implementations. In the future, computationally more efficient implementations of CryoGrid 2 could be added by means of further TILE classes which support different numerical schemes (see also Sect. 4.6).

*CryoGrid 3*: While most stratigraphy classes in the community model follow the general design of CryoGrid 3, there are some notable differences: in CryoGrid 3, the main prognostic variable integrated in time is the intensive variable temperature,

while extensive variables, such as enthalpy, are employed in the community model. This has several advantageous aspects, in particular that extensive quantities are intrinsically conserved when integrated in time. For the same reason, it is straight-





forward to combine and split grid cells when extensive variables are used as state variables. Using enthalpy as state variable is particularly convenient as both the apparent heat capacity and the thermal conductivity behave smoothly as functions of enthalpy, while especially the apparent heat capacity has a significant discontinuity at the freezing point of water when expressed
as a function of temperature. For numerical integration, this is a challenging aspect, so that using enthalpy as state variable is especially favorable for the simple time integration scheme employed in the community model. Another difference between CryoGrid 3 and the stratigraphy classes of the community model is that soil water is treated as a prognostic variable (i.e. time derivatives are calculated to advance the variable in time). In CryoGrid 3, an instantaneous bucket scheme is employed instead which computes the steady-state soil water profile after each timestep, i.e. the water profile that would eventually be reached
for the particular amounts of water added/extracted during the timestep (which means that soil water is treated as a diagnostic variable). The time dynamics of an infiltration event is therefore only captured in the community model. We emphasize that this is not a principle feature of the community model, but only applies to the stratigraphy classes described in this study. A final difference between the two models is the use of a different parametrization for the soil freezing characteristic in the community model (see Sect. 2.2.3). For most thermal simulations, we expect the impact of this change to be small (see Sect.
3.2.2 on the effect of different freezing characteristics), but the parameterization employed in the community mode in particular offers a better performance for combined thermal and hydrological simulations with Richards equation. The interested reader is referred to the discussion in Painter and Karra (2014) on this topic.

### 4.2 CryoGrid 1-3 functionalities lacking in the community model

The CryoGrid community model provides the majority of the simulation capabilities that has been demonstrated within the
CryoGrid 1 to 3 models, while at the same time providing modularity and a joint operation framework that makes additional functionality, such as the accelerated spin-up (Sect. 3.1.4), available to all model configurations. However, there are a few simulation tools published within the CryoGrid 1-3 models that are at this point not available within the CryoGrid community model. Firstly, this concerns the two-dimensional heat conduction model CryoGrid2D (Myhra et al., 2017) for which a temperature boundary condition must be specified at a number of nodes at the edge of the model domain which is not possible
with the currently available FORCING classes. Moreover, it is not directly compatible with the inherently one-dimensional stratigraphy classes described in this work. Nevertheless, the general framework of CryoGrid community model in its present form contains structures by which CryoGrid2D could be incorporated. The most direct way is to implement dedicated TILE, FORCING and OUT classes (see Fig. 1) that are geared to the requirements of two-dimensional simulations. While not directly compatible with the simulation tools presented in this work, functionalities enabled at the RUN_INFO level (Fig. 1), e.g. the
accelerated spin-up procedure, could directly be employed.

A second functionality lacking in the community model is the FLake water body model (Mironov et al., 2005) which has been demonstrated within CryoGrid 3 (Langer et al., 2016). Most notably, FLake computes a wind speed and temperature dependent stratification of the unfrozen water column, while the currently implemented stratigraphy class assumes completely mixed conditions (i.e. the entire water body has the same temperature) at all times, similar to the water body representation
in Westermann et al. (2016). It is clear that the simple representation is only adequate for shallow water bodies while FLake





must be implemented to simulate a realistic stratifications of the water column in deeper water bodies. While the simple water body representation appears to be a sufficient approximation for many permafrost applications with shallow ponds and lakes (Westermann et al., 2016), it is exactly the purpose of the modular community model that such assumptions, though seemingly logical, can be verified by comparing to simulations with a more process-rich scheme. Therefore, implementing a stratigraphy class based on FLake in the community model should become a priority for future model development.


Finally, freezing of saline ground, including diffusion of salt driven by gradients in brine concentrations (Angelopoulos et al., 2019), is a model functionality demonstrated within CryoGrid 2, but not yet included in the community model. Being a one-dimensional setup, it can be implemented in the community model as a stratigraphy class in a straight-forward way. Coupling to existing stratigraphy classes can be facilitated by dedicated interaction classes that compute the fluxes of energy, water and salt between the class interfaces. We point out that freezing of saline ground in CryoGrid 2 so far has only been demonstrated for saturated conditions and without salt advection due to water flow. Within the community model, these limitations can likely be overcome, as a stratigraphy class for saline ground can be built on top of existing stratigraphy classes with water balance representations. In a similar fashion, the existing water body class could be supplemented with a salinity representation, so that freezing of saline water bodies can be simulated. Due to the possibility to vertically connect classes, simulations of subaquatic permafrost below saline lagoons (e.g. Angelopoulos et al., 2020) and shallow near-shore zones could be accomplished within a single model framework which simulates ice formation and brine exclusion in the water column in conjunction with salt diffusion and ground freezing in the ground column below.



## 4.3 Lateral interactions

The CryoGrid community model offers a standardized way to implement lateral interactions of a one-dimensional model column with its environment, which makes it possible to account for the influence of topography and terrain heterogeneity at least in a phenomenological way. This is first and foremost useful to drive the water balance in point-scale simulations to reality. Many wetlands, for example, are located in terrain depressions and receive inflow from surrounding areas which is the very reason for their existence. Model schemes with one-dimensional water balance can not reproduce soil moisture conditions in such ecosystems, which can be a significant problem for example in carbon cycle simulations. In the CryoGrid community model, lateral inflow can be realized through a dedicated lateral interaction class which connects the model column to a water reservoir at a defined elevation. Likewise, continuous drainage of locations on slopes and hills can be simulated by using lateral interaction classes, which simulate drainage through either a seepage face or a lower-lying water reservoir. The Bayelva simulations for different drainage regimes (Fig. 14) clearly demonstrate the large impact of the drainage regime on ground temperatures, with permafrost thermally stable when water can drain, but degrading when the ground is held wet through inflow of water. In addition to the water balance, lateral fluxes of heat may play a role in permafrost areas, for example by thermally stabilizing permafrost below water bodies (Langer et al., 2016), or destabilizing permafrost in the vicinity of infrastructure elements (Schneider von Deimling et al., 2021). Similarly, a lateral interaction class can be applied in the community model to connect the CryoGrid stratigraphy to an external heat reservoir, following the same standardized way of operation as for lateral water fluxes.








A major asset of the CryoGrid 3 model is the possibility to simulate lateral interactions between different 1D model columns (denoted "tiles", corresponding to a TILE class in the community model), which facilitated simulations of complicated landscape evolution in ice-rich permafrost areas (Martin et al., 2021; Nitzbon et al., 2019, 2020, 2021). As of now, the CryoGrid community model offers the same possibilities, which will be described in detail in a future study. In short, three-dimensional lateral interaction classes are used, which rely on exactly the same structures as the one-dimensional interaction classes pre-

sented in this study. Users can therefore first conduct one-dimensional simulations coupled to a static environment, e.g. a water reservoir, which can reveal many of the effects of lateral fluxes in a semi-quantitative way. Such simulations subsequently offer an excellent starting point to set up more complicated, three-dimensional simulations with laterally coupled tiles, simply by adding information on the type of interaction and the relative positions of the different tiles to the parameter files, while the properties and model parameters related to the one-dimensional CryoGrid stratigraphies, such as ground properties and forcing

data, can remain unchanged.

### 4.4    Multi-model simulations

While evaluations of multi-model ensembles are increasingly common in cryospheric studies (e.g. van Pelt et al., 2021), most applications assume a fixed model structure. In this typical case, the evaluation of the uncertainty of simulation results is hampered by the fact that the structural uncertainty (i.e. due to imperfect model physics) tends to be lumped together with

the parameteric uncertainty (i.e. due to poorly constrained model parameters). As demonstrated in Sect. 3.2, a number of different model schemes can be efficiently simulated in the CryoGrid community model, which allows for a more thorough qualitative and quantitative assessment of model uncertainty. In this study, we have restricted ourselves to a comparison of ground temperatures simulated with different model configurations, but not attempted to identify which model configurations best explain the field measurements. In the following sections, we discuss techniques to evaluate both parameteric and structural

model uncertainty from observations which could be implemented in the CryoGrid community model in the future.

    To quantify parameteric uncertainty in a model with fixed structure, one typically assumes that the impact of unknown model parameters, including the uncertainties due to model forcing and initial conditions, can be represented by parametric error models. This is commonly achieved probabilistically through the framework of Bayesian inference (Berliner, 2003). The goal then becomes to estimate the "posterior" distribution of the parameters, which quantifies parametric uncertainty in the

model after accounting for the likelihood of the data under various plausible parameter settings (Wikle and Berliner, 2007). Since analytical forms of the posterior are not usually available, it is typically calculated approximately using numerical sampling methods (Berliner, 2003; Gelman et al., 1995). In the setting of geoscientific modeling, where each forward evaluation of the model is often computationally demanding, approximate inference is commonly performed using data assimilation (DA) methods that efficiently fuse uncertain information from physical models with observations (Carrassi et al., 2018). These meth-

ods allow physical models to be fitted to observations, while evaluating the uncertainty through a probabilistic inversion that corresponds to the first level of inference in the Bayesian hierarchy (MacKay, 2003). For cryospheric applications, ensemble-based data assimilation schemes based on the the particle and ensemble Kalman filter (Carrassi et al., 2018) have been shown to be especially promising (e.g. Aalstad et al., 2018; Fiddes et al., 2019; Alonso-González et al., 2021).



One potential issue with this kind of model fitting is that all uncertainty is attributed to the model parameters, while one effectively assumes the model structure itself to be perfect. Since all models are "wrong" to some degree (Box, 1976), this assumption is never completely justified, potentially resulting in biases in the fitted model parameters that compensate for the imperfect model. An example for this dilemma is the Bayelva simulations for different water balance schemes and drainage conditions (Figs. 14, 16): if only the bucket scheme with lateral drainage was available, one could for example adjust the strength of the lateral drainage to fit the temperature observations at the Bayelva station and conclude that permafrost is only thermally stable for well-drained conditions (Fig. 14). Likewise, one would infer from this model configuration that permafrost may have already started to degrade in areas with poor drainage for which no observations are available (the Bayelva station is located on a hill where good drainage can indeed be expected). However, the simulations with Richards equation suggest that permafrost could also be stable if no lateral drainage occurs (Fig. 16) which provides a qualitatively different assessment of permafrost stability in the wider area around the Bayelva station. In principle, these ambiguities can be resolved by considering additional observations, for example soil water contents (Boike et al., 2018). In this study, we have not explored and resolved this issue in further detail, but the example from the Bayelva area showcases a problem inherent in many cryospheric simulations.

Running multiple models with different representations of physical processes alongside one another can help to identify potential ambiguities arising from this structural uncertainty. With the CryoGrid community model, adopting such a multi-physics ensemble of models becomes relatively straightforward in practice due to the common interface (e.g. for the model forcing). Furthermore, differences in simulation results can be attributed to a specific stratigraphy class or model parametrization, as it is possible to exchange stratigraphy classes one by one and keep the rest of the stratigraphy unchanged (Sect. 3.2). In the future, the CryoGrid community model could also be used to compare different models objectively and quantitatively, which corresponds to the second level of inference in the Bayesian hierarchy (MacKay, 2003). Here, the key quantity is the "model evidence" (also called marginal likelihood) which is the probability of the observations, given a particular model. The model evidence serves as a normalizing constant when fitting a model at the first level of inference. At the second level of inference, the model evidence is used as a likelihood in Bayes' rule to update the prior probabilities of each model so as to effectively constrain model structural uncertainty using the available data. As such, the model evidence framework can be used for both Bayesian model selection and averaging. This framework is widely used in physics (Feroz et al., 2009), machine learning (MacKay, 2003), and is slowly gaining traction in geoscience (Schöniger et al., 2014). Recent work has shown that ensemble-based data assimilation schemes are also well suited for obtaining robust estimates of the model evidence (Carrassi et al., 2017). As such, a promising future avenue in CryoGrid will be the implementation of ensemble-based data assimilation both for cryospheric model fitting (e.g. Aalstad et al., 2018; Alonso-González et al., 2021; Fiddes et al., 2019) and model comparison (Carrassi et al., 2017).

The modular structure of the CryoGrid community model offers the possibility to estimate effective parameters in simplified model schemes by assimilating results from high fidelity models with richer process representations. This vision is in line with the new Earth system modeling blueprint presented by Schneider et al. (2017) which proposes using ensemble-based data assimilation to let coarse scale models learn from both observations and targeted high resolution models. For example,





transcient heat conduction-based models could be used to compile climate-dependent parameterizations for $n$-factors in the
simple TTOP model which can then be used to map large areas at a lower computational cost. In such a scheme, the simple
model is essentially used to spatially extrapolate the output of the more sophisticated model in a quasi-physical fashion. Similar
schemes could be devised to achieve long-term (e.g. multi-millennial) simulations, e.g. by parameterizing the net effect of the
surface energy balance and the seasonal snow cover so as to force the heat conduction model by monthly or even annual
averages (as in Overduin et al., 2019; Etzelmüller et al., 2020, 2022). The application of machine-learning based emulators
also opens up the possibility for full-fledged Bayesian inference of the (emulated) posterior via traditional sampling methods
which could provide a more robust and scalable framework for inverse problems (Cleary et al., 2021), thereby unifying machine
learning and data assimilation under the Bayesian framework (Geer, 2021).

### 4.5 Possible applications of the CryoGrid community model

The CryoGrid community model is designed first and foremost for the same applications as the Cryogrid 1-3 models, which
were mainly focused on the permafrost thermal regime. In addition, the community model offers the possibility to control the
lateral drainage regime of the model domain (Sects. 2.3, 4.3), a capability essential for hydrological applications and rather
limited in conventional 1D models. In particular, this allows conducting simulations with a coupled energy and water balance
for situations in which a one-dimensional representation, especially of the water balance, is not appropriate. On the one hand,
this concerns permafrost in wetlands which receive net inflow of water from surrounding areas, which also impacts the ground
thermal regime (red line Fig. 14). Although one-dimensional model configurations can not fully capture the time dynamics
of the inflow, scenario runs with different strengths of the inflow (e.g. by modifying the distance to the water reservoir, Sect.
2.3.2) could be conducted to evaluate the resilience of wetland permafrost in a changing climate in a semi-quantitative way.

On the other hand, net outflow of water typically occurs for permafrost in sloping terrain or on top of hills, which influences
both the thermal regime (blue line Fig. 14) and the water balance (Fig. 15) of the ground. This provides new applications
especially in mountain environments where the complex topography with strong elevation gradients and high spatial variabil-
ity of the surface energy balance impacts the ground thermo-hydrological regime and thus permafrost distribution (Haeberli
et al., 2010; Haeberli, 2013). Mountain permafrost has gained increasing interest in recent years, with emphasis on permafrost
degradation and its impact on rockwall stability (e.g. Gruber and Haeberli, 2007), as well as hydrology (Gao et al., 2021; Yang
et al., 2019). Previous studies have used CryoGrid 2 and 3 to simulate thermal processes in steep rock walls (Legay et al., 2021;
Magnin et al., 2017; Schmidt et al., 2021), and such applications are also possible with the CryoGrid community model. In
addition, thanks to its representations of the surface energy balance, snow cover dynamics, and lateral drainage, the CryoGrid
community model is an efficient tool to capture the effect of local topographic controls on the ground thermal regime, making
it applicable in most mountain permafrost settings.

These capacities can be combined with simulations of massive ice bodies, such as glaciers and ice patches, with a glacier class
(Sect. 2.2.8) implemented in the CryoGrid community model. In mountain environments, this offers the possibility to simulate
the complex climate-driven interactions between glaciers and permafrost, for example the development of permafrost upon
glacier retreat. While not explicitly simulating the transition, the thermal simulations for Ny-Ålesund (Sect. 3.2.2) showcase





the expected changes in the ground thermal regime. Despite the strong net ice melt (Fig. 23), the simulated mean annual temperature of the glacier is much colder than for the permafrost ground (Fig. 21), suggesting that glacier tongues at this

location at least partly feature cold-based ice, so that the area would directly transition to a permafrost setting upon glacier retreat. Combined with stratigraphy classes representing relevant ground materials (especially with excess ice, Sect. 2.2.7), the climate-driven evolution of features associated with glacier-permafrost transitions, such as debris-covered glaciers, buried glacial ice, ice-cored moraines, and ice-rich frozen debris bodies could be simulated. The rapidly changing elements of the mountain cryosphere are in particular expected to have profound impacts on stream flow, water resources, and ecosystems,

which not only affect the glacial and periglacial realms, but also adjacent valley and lowlands areas. With the possibility to jointly evaluate ground thermal regime and water balance, the CryoGrid community model can thus provide simulation capacities for a variety of climate change impact studies in mountain environments.

### 4.6 Directions and priorities for future model development

The CryoGrid community model provides an integrated platform that is designed to accelerate and simplify the development

of new simulation tools for the terrestrial cryosphere. Its modular setup provides a clear pathway for the development of new stratigraphy classes, with several defined steps that can be implemented, verified and tested independently of one another:

1. Implement the stratigraphy class itself and test it with simulations of a CryoGrid stratigraphy consisting only of this one class.

2. Implement interaction classes between the new stratigraphy class and existing stratigraphy classes that the new class shall

be compatible with; test with simulations of CryoGrid stratigraphies consisting of all supported class combinations.

3. Implement lateral interaction classes that the new stratigraphy class shall be compatible with; this can mean both the implementation of an entirely new lateral interaction class, or implementing the functionality to make existing lateral interaction classes compatible with the new stratigraphy class.

This procedure accelerates the development of complex model configurations, as developers in the first step can fully concen-

trate on the novel model physics, while the capacities of the new model functionality is amplified in the second and third steps, simply by providing defined interfaces to already existing stratigraphy and lateral interaction classes with detailed process representations. We point out that all these additions can be made without modifying the source code of the existing classes, which is important to facilitate parallel model development by several independent developers without interference. The future directions of development in this community model can therefore be left to the users and developers in a bottom-up approach.

Here, we identify a few priorities and ideas that are motivated by the results of this work (Sect. 3) and previous modeling efforts with CryoGrid 1-3. Firstly, the sensitivity experiments with different snow schemes (Sect. 3.2) show that the snow representations within CryoGrid need to be improved, as this is one of the most important sources of uncertainty in simulating the ground thermal regime (Fig. 19). With a starting point in the snow class based on the Crocus-based snow model (Vionnet et al., 2012), both simpler and more process-rich representations are conceivable. In the currently implemented Crocus-based





snow class, the snow density, which is the critical control for heat flow through the snow pack, has a strong dependence on wind speed, which can be heavily biased in model forcing data sets, especially in complex terrain (e.g. Delhasse et al., 2020). Therefore, a snow stratigraphy class in which both initial snow density and the strength of wind compaction are controlled by time-invariant empirical parameters (and not by wind speed) could be useful for many applications, as they may provide an improved performance compared to snow schemes with constant snow density (Sect. 2.2.6, Fig. 19), if the quality of the

model forcing is poor. The Crocus snow model is undergoing continuous development (e.g. Royer et al., 2021), and many improvements can be ported to CryoGrid. We see two areas in which the CryoGrid community model itself could be a useful tool to catalyze snow model improvements. First, the impact of different near-surface temperature regimes on snow properties could be investigated with relative ease; the drainage regime, for example, changes the dynamics of ground freezing during winter (Fig.14) which in turn modifies the thermal gradients and thus snow microphysics in the snow pack. Furthermore,

the snowpack on Arctic water bodies is remarkably different from surrounding land areas (e.g. Langer et al., 2016) which again can be simulated with relative ease in CryoGrid by selecting a stratigraphy class for water bodies. A comparison with observations of snowpack properties for areas with different subsurface properties can therefore reveal shortcomings of model parametrizations and eventually contribute to improvements. Secondly, explicit redistribution of snow between CryoGrid tiles due to wind drift has been demonstrated by Zweigel et al. (2021) which also has pronounced effects on the stratigraphy of

snowpack properties. While lateral interactions between tiles will be described in a later study, ablation of snow from exposed locations due to wind drift could be implemented as a lateral interaction class (Sect. 4.3) through the scheme proposed by Zweigel et al. (2021). Such simulations could again be compared to observations of snowpack properties and to the dynamics of ablation events at wind-exposed locations, such as small hills and ridges, so that both model parameters and (if necessary) the parameterizations for wind drift can be modified.

A second priority for model development is including representations for vegetated surfaces, which currently is a key shortcoming in the CryoGrid community model. A multi-layer canopy model presented in Stuenzi et al. (2021a, b) is already available as a stratigraphy class, but this is designed to represent forests and not short vegetation, such as mosses, sedges and shrubs. An intermediate-complexity representation of vegetation, e.g. following a big-leaf approach (e.g. Sellers et al., 1992), has significant potential to improve the surface energy and water balance in simulations for vegetated tundra areas. Further

development could focus on the interplay between the snow cover and different vegetation types which has the potential to improve simulations of the ground thermal regime in areas experiencing shrubification (e.g. Sturm et al., 2001). Another focus is the implementation of carbon cycling schemes of different levels of complexity, from dedicated peatland models (e.g. Chaudhary et al., 2020; Frolking et al., 2010) to novel biogeochemistry schemes with explicit representations of microbial population dynamics (e.g. Chadburn et al., 2020). Here, it is desirable to retain the modular concept of the CryoGrid community model,

so that it is possible to select and test different carbon cycle schemes that can be flexibly combined with stratigraphy classes. To be able to simulate the buildup of organic soils, stratigraphy classes must be amended with an adaptive grid which would also facilitate including sedimentation, erosion and possibly weathering in CryoGrid simulations.

At this point, most of the simulation tools in the CryoGrid community model are designed for process-scale simulations, as their computational runtime can be substantial when simulating many grid cells. A typical benchmark for simulation runtimes





(e.g. the ones in Sect. 3.2) is about 100 simulated years in 24 hours of computation (i.e. wall clock time), although this can vary strongly depending on the employed stratigraphy classes. To conduct a 1900 to 2100 climate change simulation for the entire permafrost region at 1 degree resolution would require about 100 000 to 200 000 CPU hours which is perfectly feasible on modern high performance clusters, but requires significant organizational and possibly financial efforts. It is therefore desirable to implement more efficient simulation tools, so that regional, continental and global-scale simulations become more routine
tasks.

Clustering techniques provide excellent possibilities to achieve such speed-ups by determining "typical representations", given the variability of model parameters. The simulations are then only performed for these typical representations and their output is assigned to all grid cells with similar model parameters, which is justified especially in the light of model uncertainty. Using k-means clustering, Fiddes and Gruber (2012), Fiddes et al. (2019), and Fiddes et al. (2022) achieved a runtime speedup
of three to four orders of magnitude for simulations in complex topography by clustering for terrain parameters, such as slope, aspect, and elevation. This concept could be extended to landcover maps which are one of the input layers to model-based permafrost maps (e.g. Obu et al., 2019). Furthermore, clustering could be applied to the model forcing itself, so that grid cells with similar forcing are simulated by a single representation. Such clustering would be fully compatible with the modular structure of the CryoGrid stratigraphy and could be integrated on the RUN_INFO level. With a runtime speedup of factor 100,
for example, global climate change simulations at 1 degree resolution would only require 1000 to 2000 CPU hours which would make them much more accessible to many users compared to a simulation without clustering.

Secondly, the runtime of the simulation tools could be reduced to facilitate application to many grid cells and/or over long timeframes. This can be achieved by simplifying the model physics, optimizing the model code and/or using more efficient numerical integration schemes, e.g. Runge-Kutta schemes as employed in some of the CryoGrid 2 realizations (e.g. Etzelmüller
et al., 2020). The two latter conflict with the modular setup of the CryoGrid stratigraphy in the TILE_1D_standard class, which is based on a common numerical integration scheme and standardized model structures to facilitate modularity. Therefore, we suggest adding further TILE classes (Fig. 1) custom-made to accommodate one or several efficient model tools. While these would not offer modularity and thus not be compatible with the stratigraphy classes presented in this study, there is ample opportunity for interplay and thus added value. On the one hand, faster models can obviously use parametrizations and model
parts described in Sect. 2, even if they need to be optimized for speed. On the other hand, standardized frameworks could be implemented that allow training fast models with semi-empirical elements or unresolved processes on the process-rich, but slow models of the modular CryoGrid stratigraphy (see Sect. 4.4). This could be achieved through ensemble-based data assimilation (Aalstad et al., 2018; Fiddes et al., 2019; Alonso-González et al., 2021), machine learning-based emulators (Fer et al., 2018; Dagon et al., 2020), or perhaps most promisingly a combination thereof (Bocquet et al., 2020; Brajard et al., 2020).

*Code availability.* The current version of model is available at https://github.com/CryoGrid/CryoGridCommunity_run. The exact version of the model used to produce the results in this paper, including input data and scripts to run the model, is archived on Zenodo (https:





//doi.org/10.5281/zenodo.6522424, Westermann, 2022). See Suppl. 1 for instructions for download and running the CryoGrid community model.

*Author contributions.* SW, TI and ML planned the overall concept and structure of the CryoGrid community model which was continuously

revised and extended by all authors. SW, TI, KA, RZ, CR, LS, JA, NC, JN, SS, LM, JS, ML wrote the model code. All authors contributed with testing and revising various model configurations. SW wrote the main parts of the manuscript, with contributions by all authors.

*Competing interests.* There are no competing interests.

*Acknowledgements.* This work was supported by Nunataryuk (EU grant agreement no. 773421), Permafrost4Life (Research Council of Nor-way, grant no. 301639), ESA Permafrost_CCI (https://climate.esa.int/en/projects/permafrost/), PCCH-Arctic (Research Council of Norway,

grant no. 320769), ClimaLand (EEA/EU collaboration grant between Norway and Romania 2014-2021, project code RO-NO-2019-0415, contract no. 30/2020). Jan Nitzbon, Simone M. Stuenzi, and Moritz Langer are supported through a grant by the Federal Ministry of Ed-ucation and Research (BMBF) of Germany (No. 01LN1709A, Research Group PermaRisk). Brian Groenke acknowledges the support of the Helmholtz Einstein International Berlin Research School in Data Science (HEIBRiDS). Louise S. Schmidt was funded by the Research Council of Norway through the Nansen Legacy project (NFR-276730).

The simulations were performed on resources (allocation nn9764k) provided by Sigma2 - the National Infrastructure for High Performance Computing and Data Storage in Norway, as well as resources provided by the Department of Geosciences, University of Oslo.



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
