# Peer review of "The CryoGrid community model (version 1.0) - a multi-physics toolbox for climate-driven simulations in the terrestrial cryosphere"

_Geoscientific Model Development, 2022_

## Referee Comment (RC2)

[referee-annotated manuscript omitted]

---

## Author Response (AR1)

**Author's response:**

We have compiled a revised version of our manuscript in which the comments and suggestions from two reviewers have been incorporated. In the following, we provide an outline of the major changes to our manuscript, as well as point-by-point replies to all issues raised by the two reviewers. For reviewer 2, we provide answers to the comments which we felt are critical remarks or suggestions which need to be explained and addressed in the revised version of the manuscript.

**Major changes to the manuscript:**

1. We have added a new Fig. 3, which now explicitly shows the fluxes of water and heat at the upper and lower boundary, as well as between stratigraphy classes. This figure also contains a schematic representation of the model grid within the stratigraphy classes. As a consequence, we have removed this representation of the model grid from Fig. 2

2. We have added a new Fig. 25, which shows a comparison of model runtimes for the different simulations presented in Sect. 3.2.

3. Under section 2.1.3, we have added pseudocode to explain how the upper boundary condition is handled during the CHILD-phase of the snow cover.

4. We have added a new Sect. 4.4 "Model structure and computational aspect" to the discussion which addresses several of the comments by reviewer 1. We have moved some of the text from the previous Sect. 4.6 "Directions and priorities for future model development" (now Sect. 4.7) to this new section.

**Reviewer 1:**

We thank the reviewer for constructive criticism and comments which have significantly improved our manuscript. In the following, we provide point-by-point replies to all issues raised. The reviewer comments appear in italics, our answers in normal font, and changes to the manuscript in bold.

*The paper describes the CryoGrid model, which was initially developed to study the permafrost thermal process. The current model has multiple versions, and the capabilities of*

*the model were substantially expanded, covering a broad spectrum of physical processes. The authors used Matlab's object-oriented programming to achieve modeling modularity, allowing swapping different methods and physics into the GryoGrid model. Increasing physical complexity of the model comes at the expense of computing time. It would be nice to have a graph showing how increasing model complexity affects model performance.*

-We have added a new Fig. 25 to the end of Sect. 3.2, showing the runtimes of the different simulation setups presented this section.

*The model is written in Matlab, which likely reduces the adoption of the model since not every organization or individual has access to a Matlab license. In addition, Matlab is an interpretive language and requires a certain style of code development which could lead to potential slowdowns in the execution time. I suggest including in the discussion why Matlab language was chosen and its downsides compared to the compiler languages like C++ and Fortran.*

As the earliest prototypes of the CryoGrid community model were built on the codes of the CryoGrid 3 model (Westermann et al., 2016), Matlab was mostly chosen for practical reasons, as we did not have the resources to start model development completely from scratch. While we are aware of the downsides of Matlab, as mentioned by the reviewer, we also experience some advantageous sides, in particular that the language is more easily accessible for users and developers without extensive programming skills. Secondly, the setup is standardized in Matlab, so the CryoGrid code runs on a variety of platforms without additional installations, which in our experience can discourage new users (typically again those with little programming skills). We have for example given a permafrost modeling course for a group of ten absolute beginners without any programming skills, and only asked them to have Matlab installed on their own laptops. Although there was a variety of operating systems (some old and outdated), it took us less than 20 minutes to get the model to run for all ten students. We have added a new section (Sect. 4.4) "Model structure and computational aspects" to the Discussion in which we briefly discuss limitations of Matlab, in particular with respect to runtimes. This part reads:

**Another limiting factor for runtime is the use of Matlab which is an interpreter language leading to slower runtimes. In the future, it may be possible to use automatic code translators available within Matlab ("Matlab coder") or provided by independent developers (e.g. Paulsen et al., 2016) to produce a faster C++ version of CryoGrid, in**

**particular since C++ objects feature many similarities with the Matlab classes employed in the community model.**

We have not looked at the possibility to produce C++ code in detail yet, but there seem to be no principle limitations and some initial tests looked promising. So it may be possible to come up with a C++ version of the CryoGrid community model in the future.

*Otherwise, this work is an important step that contributes to the development of the community types model. It would be nice to have a discussion about similar community-type models like CLM and others (similarities and differences).*

The main property of a community model is that it caters for applications broad enough to represent a whole science community which at the same time drives its development. In practice, there is no clear criterion when a model can be called a "community model", and many model frameworks that lack "community" in their name are in fact used and developed by a whole community. But also models which have "community" in their name have large differences, both in their organizational structure and their purpose. CLM is for example designed as land component in Earth System Models, while CryoGrid focuses on stand-alone applications. Interestingly, a certain degree of modularity has been introduced in the latest version 5.0 of CLM, while earlier versions only featured single options. As this development direction is somewhat similar to CryoGrid, we have included a discussion on this aspect in the new Section 4.4:

**In land surface modeling, such modular approaches are increasingly adopted to deal with the increasing process complexity (Fisher and Koven, 2020). As an example, the version 5.0 of the Community Land Model (CLM) offers the possibility to exchange several model components and parametrizations (Lawrence et al., 2019), in particular the soil hydrology scheme and the snow model. As CLM is designed as land component of Earth System Models, e.g. CESM2 (Danabasoglu et al., 2020) and NorESM (Seland et al., 2020), all model setups feature full land-atmosphere coupling, while "simple" schemes without surface energy balance are not provided. The CryoGrid community model, on the other hand, is more used as a stand-alone tool with broader application focus and therefore includes such simple options, in particular the TTOP equilibrium model and heat conduction models with temperature boundary condition, which are largely inherited from CryoGrid 1 and 2.**

*What are the benefits of loose versus tight coupling between different physical processes?*

The CryoGrid community model aims for loose coupling and encapsulation of the different components wherever possible. We have added a short discussion to the new Section 4.4 of the Discussion:

**The CryoGrid community model generally follows the concept of "loose coupling", keeping the degree of knowledge that each class has of the others at a minimum. Lateral interaction classes, for example, are not specifically assembled for a certain stratigraphy class, but work for a number of different stratigraphy classes. Therefore, the lateral interaction classes do not contain any code representing specific knowledge of the stratigraphy classes that they can be used with. Instead, each stratigraphy class first processes its internal information to a defined common interface format that is then passed to and processed by the lateral interaction class. This concept is applied throughout the entire model framework; while new code needs to comply with the established interfaces, it can immediately exploit much of the existing functionality. Furthermore, code additions can be made without modifying the source code of the existing classes, which is important to facilitate parallel model development by several independent developers without interference**

*Any thoughts on how the modular approach can be standardized and implemented across multiple platforms? How can mathematical programming like OpenFoam be helpful in moving toward usability and modularity?*

This is a difficult topic which we find hard to answer. In principle, we think that our approach can certainly be implemented across platforms, e.g. by using C++ objects instead of Matlab classes. The main question is what resources that would require and what organization or project would be willing to invest these resources.

We are of the opinion that providing a general discussion on the nature of community models is beyond the scope of this study. We have therefore concentrated on a few aspects which we find important pointing out, in particular the time integration scheme and the possibility to produce C++ code. We have again added them to the new Sect. 4.4 Model structure and computational aspects:

**The time integration in the CryoGrid community model (see Sect. 2.2.9) does not rely on an established partial differential equation solver (as CryoGrid 2, Westermann et al., 2013), but rather employs a simple forward Euler solver with explicit time step control. A main reason for this is to keep the code more readable, with the goal to lower the barriers for further model development, also by non-expert users. Moreover, many**

**established solver packages require the model grids to be static, and retaining full control over time integration makes it easier to include processes requiring constant changes to the model grid, such as excess ice melt and compaction of the seasonal snow cover. An obvious disadvantage of the simple time integration scheme is the relatively slow computation (see Fig. 25 for runtimes) which may be prohibitive for some applications. Another limiting factor for runtime is the use of Matlab which is an interpreter language leading to slower runtimes. In the future, it may be possible to use automatic code translators available within Matlab ("Matlab coder") or provided by independent developers (e.g. Paulsen et al., 2016) to produce a faster C++ version of CryoGrid, in particular since C++ objects are highly similar to the Matlab classes employed in the community model**

We have not mentioned OpenFOAM explicitly since this is explicitly geared towards 2 and 3-dimensional simulations, while our manuscript only concentrates on one-dimensional model setups. As mentioned before, the CryoGrid community model does can indeed be set up in 2 and 3-D configurations, but this is not covered by this manuscript. The more general aspect on "mathematical solvers" is now included in the manuscript, though.

*Minor*

*Reading this technical paper without seeing examples of the code makes the overall understanding hard. I suggest adding the snippets of the code or pseudo code to illustrate the model design and structure (e.g. Line 201, explanation of the CHILD using code).*

We have added pseudocode to Sect. 2.1.3 explaining how the upper boundary condition is computed when the snow is in the CHILD phase, as shown in Fig. 4. In addition, this section has been expanded accordingly to provide additional details in the text.

*L 345. For more clarity, it would be nice to represent the grid schematically with the boundary condition included for one or multiple stratigraphy classes.*

We have added a new Fig. 3 to show fluxes between stratigraphy classes and at the upper and lower boundary. This figure now also showcases the model grid, which have been removed from Fig. 2 to not duplicate this information.

*Are stratigraphies and physical processes the same things?*

In general, we use "stratigraphy" in the sense of "vertical distribution", which we can refer to both physical variables, such as mineral or water contents, ground temperature, etc. This is essentially the same as required in any other thermal model. In addition, it is possible to stack so-called different "stratigraphy classes" vertically (which we refer to as "CryoGrid stratigraphy", Fig. 2) which controls the representation of physical processes in the model (Sect. 2.1.1). Each stratigraphy classes can represent certain physical processes, so the user can essentially select the physical processes that she/he wants to be represented in the simulation, by selecting the appropriate stratigraphy class(es) (Sect. 2.1.2).

*L394. Soil types. Are they mixed within the bulk soil layer or discrete and vary per depth?*
A soil type represented by its Mualem van Genuchten parameters can be selected for each grid cell (i.e. by defining a stratigraphy of soil types) so that the soil type can indeed vary with depth. However, the soil type is discrete for each grid cell and mixing within a grid cell is not possible, e.g. the model does not have a representation for how to mix a grid cell with soil type "sand" and a grid cell with soil type "silt" to become "sandy silt". In this case, the user would have to manually define appropriate Mualem van genuchten parameters for sandy silt. We have clarified this in Sect. 2.2.3:

**The values of $\alpha$ and n are determined by the soil type, and users can define an unlimited number of layers with different soil types (limited by the vertical resolution of the model grid. However, only a limited number of different soil types is possible within a stratigraphy class due to the need for lookup tables which are specific for combinations of $\alpha$ and n. Currently, four soil types (sand, silt, clay, peat) are implemented to provide users with a convenient interface, but it is possible to change the $\alpha$ and n values associated with each of them, so that also other soil types can be realized.**

*2.2.9 I guess enabling different physical classes e.g. permafrost and glaciers, would significantly slow down the simulation. Can you provide any charts showing the execution time when adding more coupled different physical processes? Moving to 3D would substantially reduce the compute time. It would be nice to provide some estimates on that as well.*

Enabling different physical classes does not necessarily slow down computation, the runtime really depends on which class exactly is used. We emphasize that it is possible to restrict

time-consuming computation with process-rich stratigraphy classes to the relevant vertical domain, e.g. only computing soil hydrology for the uppermost few meters of the vertical column, while a simpler stratigraphy class is used below. This can actually lead to a somewhat reduced runtime compared to using the process-rich class for the entire domain. In the revised version of the manuscript, we provide a new Fig. 25 which shows runtimes for all model configurations displayed in Sect. 3.2. While the runtimes do vary, the spread is only about a factor of two for the configurations using the surface energy balance, and only simpler heat conduction models with temperature boundary condition or even the analytical TTOP equilibrium model have considerably shorter runtimes. The following text has been added to the end of Sect. 3.2:

**Fig. 25 displays a comparison of the runtimes of the model configurations presented in this section which are between 150 and 300 seconds per model year for most of the cases. The largest differences in runtime are caused by the boundary condition/ model type, with simulations of the surface energy balance requiring a considerably longer runtime compared to simulations in which a temperature boundary condition is applied. The shortest runtimes are achieved by the even simpler TTOP model configurations which do not simulate heat conduction at all and only compute a single equilibrium temperature. Among the model configurations using the surface energy balance as upper boundary condition, schemes with simpler model physics (e.g. constant water plus ice contents, constant snow density) have shorter runtimes than the more process-rich model configurations, but not by a large margin.**

Moving to 3D certainly increases the run-time, although the lateral interactions between tiles only make up for a small part of the runtime. The increase in runtime is mainly due to the need for synchronization between tiles, so that tiles for which computation is faster always have to wait for the slowest tile. As 3D simulations are not part of this manuscript and will be presented in more detail in future work, we do not provide estimates of runtimes for them. As a rough estimate based on experience, 3D simulations require about twice as much time as the corresponding one-dimensional simulations, when each tile is run on a separate core (as in CryoGrid 3, Nitzbon et al., 2019).

*L699. Which version of the CryoGrid is used there? Similarly, include a version of the model was used to produce figures 5 and 6.*

For the simulations, we use the class "GROUND_freeW_ubT". We have added this information in the revised manuscript.

*Figure 7. It talks about the ice impedance factor but there is no formula showing it.*

Thank you for pointing this out! The ice impedance factor is defined in Eq. 26 much earlier in the manuscript, and we have now added a cross-reference to this equation for clarity.

*Section 3.2. I suggest adding a few figures.*

We have added a new Fig. 25 on mode runtimes to this section, which now contains 16 figures. We have selected these figures carefully to showcase the capabilities of the model, in particular the possibilities to evaluate the sensitivity of model output to different parameterizations and model formulations, while avoiding to be repetitive. We fear that adding even more figures will not help the readability of this already long manuscript, and have therefore chosen to only add this one figure on runtimes.

*Figures 11 and 12 replace captions.*

Thank you, done!

*Figure 19 It looks like red and dashes red increase snow density too slowly, making soils much warmer, suggesting that these two formulations are not even appropriate.*

The simulations in red do not fit well with observations, suggesting that these are indeed not appropriate for this site. However, in our opinion this alone is not enough evidence to suggest that these schemes perform badly in general, this needs to be substantiated with simulations at further sites.

*L1108. Data assimilation or calibration?*

We mean data assimilation, e.g. using a Particle Filter, but one application could indeed be to calibrate unknown model parameters with observations, such as measured soil temperatures. In this case, one would consider an ensemble in which the parameter in question is perturbed, and the data assimilation procedure will identify the ensemble members that fit the observations best. Modern data assimilation techniques are well suited for non-linear, high-dimensional model systems, such as subsurface thermal models, which makes them in many situations superior to classic calibration methods, such as gradient descent. We have added a brief statement that data assimilation can be used for model calibration:

**These methods allow physical models to be fitted to observations (corresponding to a model calibration), …**

**Reviewer 2:**

We are grateful for the comments and suggestions provided by the reviewer which we have carefully considered when revising and improving our manuscript. In the following, we provide point-by-point replies to the key issues raised. The reviewer comments appear in italics, our answers in normal font, and changes to the manuscript in bold.

*L. 10: It is only 1D in this paper? Maybe a drawback as GryoGrid was previously used for 3D modeling.*

Yes, this manuscript only contains information on 1D model setups. We now also have 3D simulations as with CryoGrid 3 available within the community model, but this will be described in future work. We provide information on this in the Discussion section 4.3 on lateral interactions.

*L. 61: GEOtop can be listed too as an example. Also the GEOframe for modularity*

GEOtop is listed under models simulating the surface energy balance, similar to CryoGrid 3 (paragraph below). We now cite both Rigon et al., 2006 and Endrizzi et al. 2014 here.

*L. 64: delete "this"*

Done, thank you!

*L. 118: It is not adding subjectivity?*

Yes, absolutely! But it is exactly one of the goals of the modular community model that users can evaluate the impact of this choice on model results, while "normal models" with only a single configuration (in which users do not have this choice) provide only a single result, thus concealing this critical model uncertainty. A discussion of this issue and how modern data assimilation methods can be used to quantify model performance is provided in Sect. 4.5 (Sect. 4.4 in the old manuscript).

*L. 119: Idem previous models. Does not improve/simplify the parameterization.*

This is similar to pretty much all existing land surface models. It is important to point out that many model parameters (e.g. albedo) vary in space and must be set by the users, so it is not really possible to improve or simplify this from the side of the model itself.

*Fig.2: This is interesting and facilitates the development of a 3D approach.*

We agree! For 3D modeling, we also exploit the lateral interaction classes (Fig. 4) to not only connect with static reservoirs of heat and water, but with neighboring model tiles that evolve according to the surface forcing.

*Fig. 3 caption: How this threshold is defined? Again subjectivity? or can be defined based on time series of observational data?*

This is a correct observation, but many models only add the first snow grid cell when a certain SWE is reached, simply to avoid extremely thin grid cells which pose problems for the numerical scheme. Also these thresholds are user-defined and thus introduce subjectivity. For most applications with a "normal" snow cover, the CHILD phase lasts only a few days at most, so that the overall model results are not affected by the choice of the threshold. We have added the following clarification to the end of the section:

**The threshold in snow water equivalent at which the snow class becomes a normal part of the CryoGrid stratigraphy should generally be chosen small enough that the CHILD-phase does not last longer than few days, thus only having negligible effect on model results on timescales longer than a few weeks.**

*L. 272: Or the temperature offset of LST.*

We agree, this is most relevant for resolving the effects of exposition. We have added "exposition" to the factors that can be approximated by adjusting the different parameters in the TTOP equation.

*L. 480: To be realistic should be added based on the meteorological observations. -> add*

This is one possibility, others are using existing parameterizations of snow density based on the overall climatology of a site (as e.g. used in Obu et al., 2019) or by fitting observations of near-surface temperatures. As using field observations of snow density is by far most common, we have added:

**"…, which could for example be derived from field observations."**

*L. 491: How can we know that?*

This is derived from the liquid water content of the first grid cell, which is a result of the enthalpy calculations. We have added:

**with decrease rates depending on whether the snow is dry or wet (as inferred from the liquid water content of the first snow grid cell, see Sec. 2.2.3).**

*L. 548: I have no idea how this scheme is working and was previously mentioned several times.*

We have provided more details on snow compaction in Sect. 2.2.6. This section now reads:

**Snowfall is added with properties (density, grain size, dendricity and sphericity) derived from air temperature and wind speed, and the snow within each grid cell evolves and metamorphoses based on internal temperature gradient, and water content. In particular, snow compaction due to the weight of overlying snow pack is derived by computing the snow viscosity, which is computed for each snow layer, parameterized as a function of snow density, temperature, liquid water content, and snow grain size.**

*L. 594: CFL condition, Abbreviations should be explained the first time when appear*

Thank you, we have spelled it out in the revised version

On behalf of all authors,

Sebastian Westermann